## PERSPECTIVE

# Evolutionary implications of SARS-CoV-2 vaccination for the future design of vaccination strategies

Igor M. Rouzine [1,5✉] & Ganna Rozhnova [2,3,4,5✉]

Once the first SARS-CoV-2 vaccine became available, mass vaccination was the main pillar of the public health response to the COVID-19 pandemic. It was very effective in reducing hospitalizations and deaths. Here, we discuss the possibility that mass vaccination might accelerate SARS-CoV-2 evolution in antibody-binding regions compared to natural infection at the population level. Using the evidence of strong genetic variation in antibody-binding regions and taking advantage of the similarity between the envelope proteins of SARS-CoV-2 and influenza, we assume that immune selection pressure acting on these regions of the two viruses is similar. We discuss the consequences of this assumption for SARS-CoV-2 evolution in light of mathematical models developed previously for influenza. We further outline the implications of this phenomenon, if our assumptions are confirmed, for the future design of SARS-CoV-2 vaccination strategies.

The COVID-19 pandemic was a public health emergency that required massive control efforts like the 1918 influenza pandemic, the HIV pandemic, and smallpox eradication. SARS-CoV-2 variants that were substantially more transmissible or caused more severe disease than the original variant from Wuhan, China were not detected until late 2020. During this period, the public health response to COVID-19 worldwide mainly consisted of non-pharmaceutical interventions[1–5]. These measures aimed to reduce the number of transmission-effective contacts between susceptible and infectious individuals in the population and, with it, viral transmission. The interventions implemented in different settings ranged from physical distancing, mask-wearing, working from home, restrictions on public gatherings, and school closures, to bans on intercontinental travel, and complete lockdowns of entire countries[2–5].

One year into the pandemic, the non-pharmaceutical interventions were complemented by mass vaccination[6]. The quick development and rollout of vaccines around the world opened possibilities for relaxing non-pharmaceutical interventions, and it was only with the large-scale rollout of vaccination that effective control of SARS-CoV-2 transmission was achieved. However, the onset of vaccination campaigns nearly coincided with the detection of viral variants, dubbed "variants of concern" (VOCs)[7,8], that differed from the previous variants due to their demonstrated impact on transmissibility[9–11], disease severity[12–15], and the ability to evade a host's immune response[16] after natural infection or vaccination.

Despite the disparities between and within countries in access and adherence to vaccines, global vaccination coverage increased at an unprecedented pace[6]. In settings with high coverage, the main goal of mass vaccination to reduce COVID-19 hospitalizations and deaths was achieved. Given all the evidence, there appears to be little doubt that SARS-CoV-2 will continue

[1] Immunogenetics, Sechenov Institute of Evolutionary Physiology and Biochemistry of Russian Academy of Sciences, Saint-Petersburg, Russia. [2] Julius Center for Health Sciences and Primary Care, University Medical Center Utrecht, Utrecht University, Utrecht, The Netherlands. [3] BioISI – Biosystems & Integrative Sciences Institute, Faculdade de Ciências, Universidade de Lisboa, Lisboa, Portugal. [4] Center for Complex Systems Studies (CCSS), Utrecht University, Utrecht, The Netherlands. [5] These authors contributed equally: Igor M. Rouzine, Ganna Rozhnova. ✉email: igor.rouzine@iephb.ru; g.rozhnova@umcutrecht.nl

circulating as a seasonal endemic virus[17–19]. However, the seasonal pattern for SARS-CoV-2, which would be expected in the Northern and Southern hemispheres, has not settled yet and what the endemic dynamics will look like is uncertain[17,20–22]. The expectation is that the control of the virus will be achieved by vaccination alone. At this stage, the public health authorities in many countries are faced with a pressing need to make decisions on COVID-19 management and to devise future vaccination strategies for risk groups such as the elderly, healthcare workers, and individuals with medical risk conditions.

However, the pillar of the public health response to COVID-19, vaccination can also have implications for SARS-CoV-2 evolution in antibody-binding regions located in the spike protein that is targeted by the available vaccines. SARS-CoV-2 perpetually evolves due to its escape from the immune response in individuals induced by both natural infection and vaccination. Even in the absence of vaccination, there is selection pressure to escape natural immunity by accumulating mutations in T-cell epitopes and antibody-binding regions. Mass vaccination, as we show below, might increase this pressure and accelerate SARS-CoV-2 evolution in spike epitopes compared to natural infection.

In this Perspective, we first review the most important factors that shaped vaccination strategies during the pandemic. We then discuss the implications of SARS-CoV-2 vaccination on virus evolution in light of accumulated knowledge and in the context of the viral evolutionary theory. The current vaccines are designed to induce mostly a neutralizing antibody response against the spike protein, therefore we focus on the evolution of rapidly mutating antibody-binding regions. Finally, we give an outlook on further research that is needed to design potential future vaccines and vaccination strategies.

## Vaccination strategy considerations

**The importance of age.** As with other respiratory viruses such as influenza, the transmission of SARS-CoV-2 and the distribution of COVID-19 in the population are strongly influenced by several age-dependent factors[23–29] (Box 1). The factors essential for devising vaccination strategies are the number of transmission-effective contacts[23,24], susceptibility to the virus[25,26], infectivity, and severity of infections[27–29]. Respiratory viruses are transmitted via close person-to-person contact that are known to depend strongly on age[23,24]. In particular, children typically have the largest number of contacts in the population due to their interaction in schools[23,24]. This has important implications for viral transmission. For example, children are believed to have played a large role in the spread of the pandemic influenza A H1N1 in 2009 due to their generally high number of contacts and high susceptibility to the virus[30,31]. The age-dependent contact patterns relevant to SARS-CoV-2 transmission are similar. However, unlike influenza, the susceptibility to SARS-CoV-2 increases with age[25,26,32,33]. Children and adolescents younger than 20 years are estimated to be about 50% less susceptible to

---

**Box 1 | Vaccination strategy considerations**

**Biological characteristics of infection**

- Infectiousness during the infectious period
- Duration of the latent period
- Duration of the infectious period
- Probability of transmission per contact between a susceptible and an infectious individual

**Importance of age**

- Number of transmission-effective contacts between susceptible and infectious individuals
- Susceptibility to infection
- Infectivity of infections (asymptomatic, symptomatic, and severe disease)
- Disease severity (asymptomatic, symptomatic, severe disease, hospitalization, death, etc.)
- Duration of immunity after natural infection or vaccination

**Properties and performance of vaccines**

- Protection against disease, hospitalization, and death
- Protection against any infection (infection-blocking property)
- Indirect protection due to reduced infectivity of breakthrough infections

**Setting-specific factors**

- Level of non-pharmaceutical interventions
- Vaccine supply and vaccine hesitancy
- Seroprevalence prior to vaccination rollout
- Ethical, legal, political, and practical considerations

**Vaccination objectives**
The immediate goal of the WHO's global COVID-19 vaccination strategy 9

- Minimize deaths, severe disease, and overall disease burden
- Curtail the health system impact
- Fully resume socio-economic activity
- Reduce the risk of new variants

The updated goal of the WHO's global COVID-19 vaccination strategy 15:

- Sustain efforts to reduce mortality and morbidity, protect the health systems, and resume socio-economic activities with existing vaccines
- Accelerate development and access to improved vaccines to achieve durable, broadly protective immunity, and reduce transmission

infection than adults who are 20 years and older, and very young children even less[26,34]. In addition, SARS-CoV-2 disease severity also increases rapidly with age[27–29,32,33]. Young children harbor a relatively large proportion of sub-clinical asymptomatic infections[27]. These factors, coupled with the evidence of lower infectivity of asymptomatic infections, compared to the infectivity of severe infections, resulted in children having a rather modest contribution to transmission and a very low share of severe disease in the population during the waves of the wild-type variant and of early VOCs[28,29,32,33]. During the waves of Delta and Omicron BA.1/2, children suffered a larger burden of infections than before but these infections still rarely resulted in severe morbidity and mortality relative to older age groups[35]. On the other side of the age spectrum, the elderly have the highest probability of clinical disease, hospitalization, and death, the last two increasing exponentially with age[28,29,32,33]. This evidence factors in heavily when public health authorities have to make decisions on which groups in the population should be prioritized for vaccination.

**Properties and performance of vaccines.** There are several different SARS-CoV-2 vaccine formulations including mRNA vaccines (with a lipid envelope and a protein subunit) and vector vaccines. The main goal of all types of vaccines is to induce high levels of neutralizing antibodies mimicking the response after natural infection. Regardless of their formulation, all vaccines approved by major public health agencies were shown to have three main protection effects (reviewed in ref. [36]) that are important for devising vaccination strategies (Box 1). Firstly, vaccines offer a consistently high level of protection against clinical disease, hospitalization, and death in vaccinated individuals in both randomized clinical trials and real-world effectiveness studies. Secondly, vaccines have an indirect protective effect due to the reduced infectivity of breakthrough infections in vaccinated individuals relative to natural infections in unvaccinated individuals. Thirdly, vaccines demonstrate infection-blocking properties whereby they protect vaccinated individuals against any, even sub-clinical, infection. While the degree of protection depends on the vaccine brand, genetic factors, medical risk conditions, state of the immune system, the age of a vaccinated individual, and virus variant, these general protection effects are observed universally across all vaccine types and all ages. It is important to stress that, while SARS-CoV-2 vaccines offer some protection against infection, however high this protection may be, it is never perfect, i.e., the efficacy against infection is below 100%. For SARS-CoV-2, the efficacy against infection is generally lower than the efficacy against hospitalization[36]. The same holds true for influenza vaccines, for which vaccine effectiveness against infection is estimated at 30–60%[37]. In contrast, vaccine protection against infection with the yellow fever virus and most other childhood infections like measles-mumps-rubella exceeds 99%[38]. The vaccine efficacy or real-world effectiveness against the VOCs generally declined when compared with the Wuhan variant[36]. The biggest reduction was in the efficacy against infection while the efficacy against hospitalization and death were reduced much less. The decline in vaccine efficacy to decrease the number of infections was partly because the original vaccines were designed specifically for the initial virus, and, partly, due to the immune escape detailed below. At the moment, we cannot quantify which effect is larger. Bivalent mRNA vaccines adapted to target the original variant and a more recent Omicron subvariant have now been approved and deployed to a limited extent, similar to influenza vaccines that are matched to the most prevalent circulating variant.

**Vaccination objectives.** The public health objectives of vaccination are defined by policymakers. The choice of a vaccination strategy requires a clear definition of the outcomes that this strategy aims to achieve. In different situations and for different pathogens, some of the most desirable outcomes of vaccination may be preventing or delaying a pandemic, reducing the peak of infections or hospitalizations, minimizing the duration of a pandemic or the total number of individuals in the population that will become eventually infected before it ends. Regarding COVID-19 (Box 1), the immediate concern of policymakers was to keep the healthcare systems functioning[39]. In practice, this means that the peak of hospitalizations must not exceed the healthcare system's capacity. Another concern was reducing the total mortality due to COVID-19[39]. The situation regarding minimizing the total number of infections differed by country. Some countries like China applied a zero COVID-19 policy that aimed to keep COVID-19 cases as close to zero as possible by imposing very strict public health measures. In other countries, including the Netherlands, the consequences of SARS-CoV-2 infection without hospitalization, such as long COVID and work absenteeism, were recognized, but still, there was no goal to minimize the total number of infections. The vaccination objective as updated in June 2022 (Box 1) recognized the need to sustain efforts to reduce mortality, morbidity, and transmission by expanding vaccination among those at greatest risk (healthcare workers, people over 60 years old, and other at-risk groups)[40]. With these public health objectives in mind, we can look through all possible vaccination strategies to determine the most desirable and practically achievable result.

**Vaccination strategies.** The best tools for devising control measures during the current pandemic were predictions based on robust epidemiological models calibrated to the available data[1,32,33,41–43]. These mathematical models known as "dynamic transmission models" or "infectious disease models" simulate the transmission of a virus in a population and allow the evaluation of potential control measures using computers. Transmission models were used to devise vaccination strategies and served as a basis for policymaking in many countries[33,41–43]. The model-based assessment of the vaccination impact incorporates biological characteristics of the virus, age-dependent effects in transmission and disease severity, epidemiological effects of vaccines, and additional factors that are setting-specific (e.g., the level of non-pharmaceutical interventions, vaccine supply, and SARS-CoV-2 seroprevalence prior to vaccination rollout[29]) (Box 1). Following recommendations based on these models and considering a number of practicalities, vaccines were rolled out in a similar fashion in many countries. As a general rule, several vaccination groups (e.g., staff and residents of long-term care facilities, healthcare workers, and individuals with comorbidities) were prioritized for vaccination. After this, the vaccination rollout proceeded by eligible age cohorts, starting from the oldest and ending with the youngest[39], closely following the criteria of risk for COVID-19 hospitalization and death. Such targeted control measures are common in epidemiology. More recently, the vaccination strategy focused on protecting vulnerable groups such as the elderly, healthcare workers, and individuals with medical risk conditions[40].

**Global vaccination rollout.** Despite the disparities between and within countries in the access and adherence to vaccines, the global rollout has been extraordinary in terms of speed and magnitude[6]. According to recent estimates, only in a subset of 33 countries in the World Health Organization (WHO) European Region the widespread implementation of COVID-19 vaccination

programmes averted almost half a million deaths in people 60 years and older[44], and almost fifteen million deaths were prevented by vaccination in 185 countries and territories between December 8, 2020 and December 8, 2021[45]. The desire of authorities to reduce viral transmission, protect the population from COVID-19, and speed up the return to normality, turned into a vaccination race aimed at vaccinating all eligible individuals as fast as possible. Many countries were successful in achieving high vaccination coverage, up to 90–100% in the oldest age categories[6]. It is important to note that, because vaccines were rolled out during the ongoing pandemic, the vaccination strategy had to be adapted over time in response to new VOCs and declining protection after natural infection and vaccination. Other factors that started to play a role in the assessment of vaccination strategies were the duration of immune protection after natural infection or vaccination, as well as the cross-protection from prior infection with one variant against another variant or other seasonal human coronaviruses[16]. A booster vaccination is seen as a way to increase the protection of vaccinated individuals against the new VOCs and to keep COVID-19 hospitalizations and deaths at bay in those at greatest risk.

## Evolutionary consequences of vaccination at the population level

Similarly to HIV, HCV, and influenza virus, SARS-CoV-2 is perpetually acquiring new mutations in its genome, with an average substitution rate of $(0.7\text{-}1.1) \times 10^{-3}$/year/site[46] (Fig. 1). SARS-CoV-2 evolution is especially fast in the spike protein[47–50]. Three major reasons account for the rapid evolution in viral receptor proteins, such as the spike of SARS-CoV-2, hemagglutinin of influenza, and gp120 of HIV. Firstly, the spike has receptor-binding motifs that affect the transmission, and their evolution leads to an increase in virus fitness. This may play a role in the emergence of VOCs with enhanced transmissibility[9–11,51]. Secondly, it contains epitopes, regions that are very important for the immune response because of their involvement in the binding of antibodies that can neutralize the virus. Mutations in epitopes, in addition to the waning of antibodies, are a major factor that limits viral recognition by the immune system and, hence, the durability of protection against infection[47,52,53]. Thirdly, these epitope regions have evolved to have low physiological constraints on mutation (low mutation cost) because they serve primarily as highly-variable decoys for antibodies. Had they another important function for a virus, they would be conserved. For example, the receptor-binding site has a function and is conserved, because it hides between the protruding variable regions to prevent antibody binding. This is the case for both the influenza virus receptor (hemagglutinin) and HIV receptor (gp120)[54–57].

In the next sections, we focus on the population-level evolution in neutralizing antibody epitopes related to immunity, vaccination, and viral recognition and on escape mutations that occur during transmission chains of acute infections. We postpone until the final section the discussion of the evolution outside of epitopes and of rare chronic infections, where escape mutations can accumulate within one individual.

In what follows, we assume—and this is the only essential assumption to be tested in the future experiments on which our discussion relies—that for SARS-CoV-2 the cost of mutations in antibody-neutralizing regions to virus replication ability is as small as that for influenza virus and HIV. We make this assumption because the structure of antibody-binding sites on the spike protein of SARS-CoV-2 is similar to the structures on gp120 of HIV and hemagglutinin of influenza. It comprises several protrusions of similar lengths covered in sugars, located far from

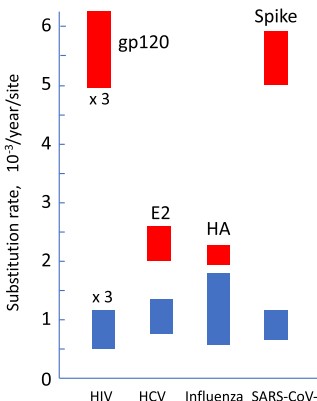

**Fig. 1 Viral substitution rates.** Blue rectangles show the intervals of the median values for the most rapidly and most slowly evolving subtypes of HIV, HCV, influenza virus, and SARS-CoV-2 for the full genome[46,161–163]. Red rectangles show the rates for the proteins targeted by neutralizing antibodies[47,54,164,165]. HIV data are multiplied by 3.

the receptor-binding site, and serving as targets for antibodies. Then, the selection pressure for these viruses to escape is reasonably expected to be of the same order of magnitude. In the general case, the final cost of mutation limits the antigenic escape[58–60], so this assumption remains to be tested in the future experimentally. Most research has, so far, focused on mutations causing the emergence of VOC, and we hope that this discussion will bring the focus to finding new epitope variants for SARS-CoV-2 as it happened for influenza and HIV.

The maximal vaccination coverage supported by public health authorities for controlling the pandemic had important consequences. In the shorter term of a few months, this approach facilitated a decrease in the number of infections, helped to unload hospitals, and to reduce COVID-19-related mortality. In the longer term, as we show in this Perspective, mass vaccination potentially further accelerates the rapid evolution of epitope regions. Let us start, however, with unvaccinated populations.

## Virus evolution in epitopes in the absence of vaccination: the Red Queen effect.

In the absence of vaccination, virus evolution in epitopes at the population level is driven by the immune response in individuals recovered from natural infections whose number gradually accumulates in the population. The process is observed for other respiratory viruses, including influenza, and it has been investigated in detail in genomic, immunological, and bioinformatic studies[61–67]. Its evolutionary dynamics has been interpreted and predicted using mathematical models[54,68–75]. The perpetual immune escape in epitopes is the reason why seasonal influenza is not becoming extinct but persists among humans. In order to avoid extinction, the influenza virus has to continue mutating to distance itself genetically from the immune response accumulating in the population. Such adaptation of an organism facing an evolving opposing species is termed "The Red Queen effect" (Fig. 2).

The persistent epitope evolution is observed in seasonal human coronaviruses and SARS-CoV-2[47–50]. However, the substitution rate in the spike protein of SARS-CoV-2, which is three times faster than that of influenza's hemagglutinin, is unprecedented for an acute respiratory virus[47]. The consequences of this process become clear if we introduce two quantities describing the potential of a virus to spread in a population, the basic and effective reproduction numbers, $R_0$ and $R_e$ (Box 2). Both measure the Darwinian viral fitness on the population level defined as the number of individuals infected by one individual. They

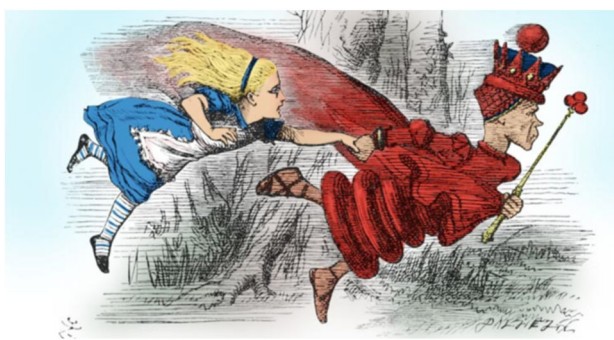

**Fig. 2 The Red Queen effect.** The effect bears the name of the Red Queen's race from the novel "Through the Looking-Glass" by Lewis Carroll. As the Red Queen told Alice: "Now, here, you see, it takes all the running you can do, to keep in the same place." Similarly, the evolution of SARS-CoV-2 in epitopes at the population level is driven by the immune response in individuals recovered from natural infection or vaccinated, whose number gradually accumulates in the population. In order to avoid extinction, the virus has to mutate perpetually to distance itself genetically from this immune response accumulating in the population. Image source: The picture is a modified version of the illustration by Sir John Tenniel from Lewis Carroll's "Through the Looking-Glass", 1871, downloaded from https://www.alice-in-wonderland.net/resources/pictures/through-the-looking-glass/.

incorporate several host-level factors, including the number of virions per cell, infectivity, transmission dose, the immune response in a host, and transmission bottlenecks. The former, $R_0$, denotes the average number of newly infected individuals caused by one infected individual in a fully susceptible (naïve) population. The latter, $R_e$, denotes the average number of new infected individuals caused by one infected individual in a population where some individuals are immune. Both $R_0$ and $R_e$, are defined for given public measures in place, such as lockdowns and various restrictions. They differ, by definition, only due to the immune memory accumulated in a population. Therefore, public health measures reduce $R_0$ and $R_e$ by the same factor.

$R_0$ is larger than 1 for the original SARS-CoV-2 variant (range 1.9 – 6.5)[76] and for Alpha, Delta, Gamma, Omicron and other VOCs[8,9,11,77]. This condition means that each infected individual transmits the virus to more than one other individual which initially leads to an exponentially increasing number of infected individuals. With more and more individuals being infected and becoming immune, $R_e$ would keep decreasing until, with no other changes, transmission would become negligible.

However, as a result of the ongoing evolutionary escape from the immune response after natural infection, as well as due to mutations outside of the epitopes including the receptor-binding domain, this outcome is altered to the stationary process with seasonal oscillations in the virus prevalence due to seasonality in transmission. At the beginning of each seasonal epidemic, $R_e$ is larger than 1. As the virus infects more people who acquire immunity, $R_e$ becomes lower than 1. When averaged over seasonal epidemics of several years, $R_e$ is close to 1. Thus, each infected individual transmits the virus, on average, to one individual keeping the number of the infected individuals averaged over a long period of time approximately constant[73–75]. This stationary process with seasonal oscillations is established after most people have already been infected at least once.

**Models of virus evolution in epitopes at the population level.** Mathematical models connect the initial assumptions to predictions in the most accurate and reproducible way. A key benefit of using models is that they allow uncertainty to be quantified and to conduct scenario analyses based on a range of initial assumptions. An important observable quantity predicted by mathematical models and measured from viral genome sequence data is the speed of antigenic evolution or substitution rate, $V$, defined as the rate of accumulation of non-synonymous mutations in neutralizing antibody epitopes (Box 2). Using mathematical models, $V$ can be expressed in terms of several parameters, namely the number of infected individuals, $N_{inf}$, and mutation rate, $U_b$, defined as the probability of an escape mutation per transmission in epitopes (Box 2). The latter is a composite parameter that depends on the conditions in an individual host including the virus replication error rate in a cell and the host immune response creating natural selection for immune escape mutants[78,79]. Substitution rate, $V$, also depends on the virus recombination rate and the transmission advantage of an escape mutant, $s$. The form of the relationship between $V$ and parameters $N_{inf}$, $U_b$, $s$ may vary depending on the most important factors of evolution and the values of these parameters.

In the simplest case, when the product $N_{inf} U_b$ is much less than 1, immune escape mutations emerge and spread through the population one at a time (Fig. 3a). In this case, the substitution rate, $V$, is proportional to $N_{inf} U_b$. This assumption is implicitly built into some epidemiological models of seasonal influenza and SARS-CoV-2[78,80–82].

However, if the product $N_{inf} U_b$ is much larger than 1, mathematical models predict that mutations occurring at different positions of the viral genome are concurrent in time, and the approximation of independent sweeps does not apply anymore due to strong interference between different mutations[83–92]. This linkage interference creates several effects such as interference between emerging clones (Fisher-Müller effect)[93,94], which is equivalent[95] to Hill–Robertson effect[96], i.e., the decrease of selection effect at one locus due to selection at another locus, as well as various genetic background effects. All of these effects are directly observed for seasonal influenza and many other viruses[54,93,97,98] (Fig. 3b). The linkage interference effects slow down virus evolution by orders of magnitude and change the way the substitution rate[83–92], $V$, and the statistics of phylogenetic trees[99,100] depend on the number of infected individuals, $N_{inf}$, mutation rate, $U_b$, and recombination rate. We refer to this situation as "multi-locus regime."

SARS-CoV-2 genome has hundreds of evolving sites[101]. The virus demonstrates, on average, more than two substitutions per month, or $1.1 \times 10^{-3}$ substitutions per site per year[46], and modest intra-host diversity[102,103]. Fast evolution is common for RNA viruses because they lack proofreading enzymes, their mutation rates are relatively large[104]. They all fall in the range of $10^{-6}$ to $10^{-4}$ per nucleotide per replication. To estimate $U_b$, we have to multiply this mutation rate by the size of the infected population and the length of the antibody binding region. For influenza A, the mutation rate per transmission event per antibody binding region is estimated at $3 \times 10^{-4}$ [64,73]. For SARS-CoV-2, the population-level mutation rate is expected to fall within the same order of magnitude. Therefore, the multi-locus regime ($N_{inf}\ U_b > 1$) applies if more than $N_{inf} = 10,000$ infected individuals are present in a population, which is the case during a pandemic wave in a large city. The fact that $N_{inf}\ U_b > 1$ for influenza A H2N3, which falls into the multi-locus regime, was demonstrated using sequence data[54].

Thus, SARS-CoV-2 evolution can be described by multi-locus models, which have been studied intensely over the last two decades using the methods of statistical physics[83–92,99,100]. Their exact predictions for the substitution rate, $V$, vary depending on the specific evolutionary factors taken into consideration. Genetic variants arise by random mutation but are subsequently amplified

---

**Box 2 | Glossary**

**Epidemiology**

- **Basic reproduction number (or basic viral fitness), $R_0$**: Average number of newly infected individuals caused by one infected individual in a fully susceptible (naïve) population.
- **Reproduction number (or viral fitness), $R_e$**: Average number of newly infected individuals caused by one infected individual in a population where some individuals are immune.
- **Vaccine efficacy**: Percentage reduction of the numbers of infections, cases of severe disease, hospitalizations, and deaths in a vaccinated group relative to an unvaccinated group in a randomized clinical trial.
- **Vaccine effectiveness**: Same as the vaccine efficacy but determined in real-world studies.
- **Virulence**: Ability of a virus to cause severe disease and, in particular, death upon infection.

**Immunology**

- **Epitope**: A segment of the viral protein recognized by the host immune system.
- **Immune escape mutation**: Mutation decreasing viral recognition by the immune system.
- **Cross-reactivity half-distance**: The number of amino acid substitutions in epitopes such that the host susceptibility to the virus is half of its maximum value measured in a fully susceptible (naïve) individual.

**Evolution**

- **Average effective selection coefficient, $s$**: Average relative change in $R_e$ due to mutation.
- **Beneficial mutation**: Mutation increasing $R_e$.
- **Clonal interference**: Competition for human hosts between two concurrent in time, beneficial mutations occurring at two different positions of the viral genome.
- **Defective interfering particle**: A replication-incompetent virus that uses proteins of the wild-type virus to replicate.
- **Effective mutation rate, $U_b$**: Probability of mutation per transmission event per nucleotide.
- **Effective outcrossing number**: Probability of recombination with another viral variant per transmission cycle.
- **Fitness landscape**: Dependence of $R_e$ on the genetic sequence.
- **Genetic distance, $x$**: Number of nucleotide differences between the genomes of two viral variants.
- **Non-synonymous mutation**: Mutation that changes an amino acid.
- **Recombination**: Fusion of the genetic material of two viral variants into a new variant.
- **Substitution**: Mutation replacing a nucleotide with another.
- **Selection pressure**: Increase in $R_e$ due to mutation.
- **Substitution rate, $V$**: Accumulation rate of mutated nucleotides, referred to as the speed of evolution in epitopes.

---

or suppressed by natural selection, with random genetic drift and linkage as additional stochastic factors. However, all these models demonstrate that $V$ depends weakly on both the number of currently infected individuals, $N_{inf}$, and mutation rate, $U_b$. More specifically, $V$ is linearly proportional to $\log N_{inf}$ and increases logarithmically with $U_b$ as well (Fig. 3b).

The multi-locus models predict that the average substitution rate, $V$, is proportional to the selection pressure that describes the change of viral fitness due to mutation[83–88,90,92]. The selection pressure in epitopes due to immunity accumulating in an unvaccinated host population has been investigated in several modeling studies[73–75]. These models account for the immune response after natural infection, and their most important result is the general expression for the average transmission advantage of an escape mutation in an epitope created by natural immunity in recovered individuals. This quantity, analogous to the average selection coefficient in evolutionary theory, $s$, is expressed in terms of only two viral parameters, one epidemiological and another immunological. The epidemiological parameter is $R_0$, which we introduced earlier. The immunological parameter is the cross-immunity distance, $a$, defined as the value of the genetic distance between the infecting virus and the virus from which the individual recovered previously, at which the host susceptibility to the virus is half of its maximum value in the fully susceptible (naïve) individual. The cross-immunity distance, $a$, is a composite parameter that combines the cross-reactivity of antibodies with their total number left after infection. The average transmission advantage of an escape mutation in an epitope has the general form[73–75]

$$s = \frac{1}{a} f(R_0) \qquad (1)$$

where the function $f$ grows slower than a linear function, and $f(1) = 0$. Equation 1 means that the average transmission advantage of an escape mutation in an epitope, $s$, is lower for a larger cross-immunity distance, $a$, i.e., when antibodies are more broadly neutralizing (epitope binding by antibodies is less sensitive to mutations in epitope). Equation 1 also states that $s$ grows with the basic reproduction number, $R_0$. Indeed, $R_0$ defines the viral transmission potential in the naïve population and, hence, sets the scale for transmission changes with mutation.

The specific form of function $f$ in Eq. 1 depends on the properties of the natural immune memory and cross-recognition, and varies among analytic multi-locus models[73–75]. While these studies differ in the description of natural immunity and, hence, in the accuracy of the predictions, they all agree on two important predictions. Firstly, these studies produce expressions for the selection pressure driving the antigenic wave in the form of Eq. 1. Secondly, corroborating earlier numerical findings[69,70], the evolution of the virus and immune memory in a population is predicted to be a traveling wave in the genetic space that has a quasi-one-dimensional shape.

Cited studies[73–75] assumed that the genetic distance is linearly proportional to antigenic distance. An additional complication is that different mutations in epitopes have a variable effect on epitope binding. Some mutations make a large change in the free energy of binding, and some mutations are almost neutral. This means that the genetic distance and antigenic distance are not linearly proportional. In terms of the phenotype landscape, it has a rugged component. This fact has been investigated in detail using two-dimensional antigenic maps for influenza[61,70] and is of major importance when short-term evolution is studied. In the

**Fig. 3 Schematics of the dependence of the population-level viral substitution rate on the proportion of the immune population.** Immune population consists of individuals who recovered from natural infection and who were vaccinated. Using mathematical models, population-level viral substitution rate in epitopes, $V$, can be expressed in terms of the effective number of infected individuals, $N_{inf}$, and mutation rate, $U_b$, defined as the probability of an escape mutation per transmission in epitopes. **a** At a low mutation rate per population per epitope per transmission, $N_{inf}U_b \ll 1$, the supply of immune escape mutations is low. As a consequence, escape mutations spread in the population one at a time. The population-level substitution rate $V$ (black line) is proportional to the immune selection pressure in a population $s_{tot}$ (gray line) and the total number of infected individuals where escape mutants can emerge $N_{inf}$ (red line). The dependence of $V$ on the proportion of the immune population has a maximum (black line). **b** At a high mutation rate, $N_{inf}U_b \gg 1$, many escape mutations at different positions of the viral genome spread in the population at almost the same time and compete with each other for human hosts. The substitution rate $V$ (black line) is proportional to selection pressure $s_{tot}$ (gray line) and weakly depends on $N_{inf}$ (red line), so that $V$ increases monotonically with the proportion of the immune population. **a**, **b** Blue people are recovered; orange, yellow, and red people are infected with viral variants with different immune escape mutations (genomes below). The evolution of seasonal influenza and SARS-CoV-2 is compatible with **b** and not with **a** (see text).

long-term evolution, an effective ratio of antigenic to genetic change can be successfully described by studies with the averaged-out cross-immunity function[71,73–75].

**Effect of vaccination on viral evolution in epitopes at the population level.** Vaccination slows down transmission by decreasing the probability of infection of vaccinated individuals and by decreasing the viral load in such individuals should they become infected. Put simply, vaccination also makes individuals less infectious. Some studies argued that, because the virus is poorly transmitted in the vaccinated population, the reduction in the effective reproduction number by vaccination will also slow down the evolution of epitopes[78]. This prediction would be correct under the assumption that $R_e$ was reduced below 1 very rapidly and remained at that level for a period of time long enough for the virus to be eradicated. The situation is analogous to the successful antiretroviral therapy in an HIV-infected individual. The therapy not only suppresses the number of infected cells by several orders of magnitude, but also strongly impedes within-host evolution, because it reduces the initial reproduction number below 1 very rapidly. An example at the population level is vaccination against childhood infections like measles-mumps-rubella which has a very high efficacy against infection and eradicates the virus in a population with sufficiently high vaccination coverage.

Unfortunately, due to the combination of factors such as vaccine efficacy against infection below 100%, incomplete vaccination coverage, mutations in the receptor-binding region, and pre-existing epitope mutants, $R_e$ of SARS-CoV-2 does not fall fast enough. It soon rebounds above 1 allowing for the virus evolution in a population to continue. In situations when

vaccination does not have the rapid eradicating effect, it does not slow down but, on the contrary, applies additional selection pressure due to the additional immune memory cells it creates, similarly to the selection pressure from natural infection[70,73–75]. If the cost of mutations is low enough, this immune selection pressure will further accelerate virus evolution in antibody-binding regions[58–60]. The effect is analogous to the case of suboptimal therapy in an HIV-infected individual that selects drug-resistant mutants. These mutants exist in very small quantities before therapy and become dominant in a patient within weeks of failing therapy. Highly-active drug cocktails have solved this problem. A similar dichotomy for neutralizing and non-neutralizing vaccines was predicted for the evolution of virulence[105]. Note that since we consider the dynamics at the population level, the effect discussed here will be the same both for a "leaky" vaccine, where all susceptible individuals have reduced susceptibility to infection after vaccination, and for an "all or nothing" vaccine, where a proportion of susceptible individuals are completely protected by vaccination.

In the case of the epitope evolution of SARS-CoV-2 in a population, vaccination adds the immunity of vaccinated individuals to the natural immunity of recovered individuals, thus potentially favoring immune escape mutations even more. Using an equestrian analogy, vaccination spurs the Red Queen into a full gallop. Thus, the cost of the short-term decrease in the number of infections is the spread of new epitope mutations in the future. The extent to which vaccination accelerates evolution in antibody-binding regions depends on vaccination frequency, inter-dose period, molecular design of vaccines, and details of immunological and evolutionary dynamics. To avoid confusion, we emphasize that we discuss here the evolution of epitopes only

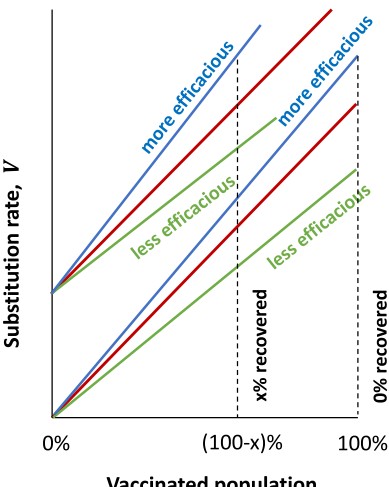

**Fig. 4 Schematics of the dependence of the substitution rate on the proportions of vaccinated and recovered population and on the relative efficacy of vaccine compared to the natural immune response.** The substitution rate in epitopes is shown in the presence (three higher lines) and in the absence (three lower lines) of recovered individuals as the proportion of the vaccinated population increases. Red, green, and blue lines correspond to vaccine efficacy in inducing protection against infection equal, lower, and higher than the natural immune response. We assume the absence of fully susceptible (naïve) individuals.

and not, for example, the emergence of VOC related to mutations in other regions. Before Omicron and its descendants, all VOCs had emerged before the mass rollout of vaccination as inferred from phylogenetic analyses[106,107]. Vaccination was probably not involved as a selection pressure in their genesis.

Just to illustrate the potential magnitude of the Red Queen effect, let us make a ballpark estimate. In the case of high vaccination coverage combined with non-pharmaceutical interventions, we observe a sharp short-term drop in the total number of currently infected individuals, $N_{\mathrm{inf}}$. The long-term impact of the decrease in the number of infected individuals on viral evolution in epitopes can be found from the substitution rate, $V$. As already mentioned, all multi-locus models demonstrate that $V$ depends logarithmically on both $N_{\mathrm{inf}}$ and mutation rate, $U_b$[108].

For example, if vaccination has decreased the number of infected individuals in a city 100-fold from 10,000 to 100, the corresponding decrease in the virus substitution rate is only two-fold. This is a drastically different result from a 100-fold reduction expected from the simpler models assuming rare immune escape mutations where the substitution rate is linearly proportional to product $N_{\mathrm{inf}} U_b$. The same consideration applies to parameter $U_b$ which has been argued to depend on the vaccination dose in a vaccinated host[78,79]. The effect of a change in $U_b$ on $V$ is very small (i.e., logarithmic). At the same time, the substitution rate, $V$, is linearly proportional to the selection pressure, $s$, created by the individuals with natural or vaccine-induced immunity[73–75], Eq. 1 (Fig. 4).

Thus, vaccination creates two opposing effects on the Red Queen adaptation in epitopes. One effect comes from the decrease in transmission rate due to partial immune protection and lower virus amount transmitted to another individual. This effect creates a positive selection pressure for resistant mutations. The opposite effect of vaccination comes from the decrease in the mutation rate within a host, due to lower viral load. The first effect wins, because the adaptation rate is linearly proportional to selection pressure, $s$, and weakly depends on the mutation rate, $U_b$ (Fig. 3b).

Several studies[78,80,81] use standard compartmental infectious disease models and arguments following those to predict that vaccination can decrease vaccine escape by reducing the number of infectious individuals. We would like to point out that the aforementioned studies lack evolutionary dynamics because it is not built into these simplified models. For example, ref. [80] assume the vaccine escape pressure to be proportional to the number of vaccinated individuals. At best, models of this type correspond to the case when immune escape mutations emerge and spread through the population one at a time (single-site approximation or independent-locus models; Fig. 3a). As we explain above, this is the case for unrealistically-small population sizes. These models do not include proper treatment of evolutionary dynamics and disregard linkage effects existing between mutations at multiple sites, because such models are technically more difficult to handle. Here, we use the modern theory of multi-locus virus evolution that takes into account clonal interference, genetic background effect, and other linkage effects, random genetic drift, and natural selection arising due to immune memory. From the evolutionary viewpoint, multi-locus models are closer to reality than independent-locus models. At the same time, based on strong genetic variation in the antibody epitopes in the spike protein of SARS-CoV-2, we assumed that mutations in these regions have a low cost, by analogy with mutations in hemagglutinin of influenza. The validity of these assumptions remains to be tested directly in the future.

**Estimation of the vaccination effect on the substitution rate in epitopes.** The modeling results obtained for natural immunity[73–75] can be generalized for immunity induced by vaccination. Due to the immune response in vaccinated individuals the total selection pressure of escape, denoted $s_{\mathrm{tot}}$, is increased by an additional term, denoted $s_{\mathrm{vac}}$, as follows

$$s_{\mathrm{tot}} = s + s_{\mathrm{vac}} \qquad (2)$$

where $s$ is given by Eq. 1, and $s_{\mathrm{vac}}$ corresponds to the effect of vaccination averaged over time. Because the substitution rate, $V$, is linearly proportional to the selection pressure, Eq. 1, it increases linearly with the vaccination term, $s_{\mathrm{vac}}$. The selection pressure due to vaccination, $s_{\mathrm{vac}}$, depends on the type of vaccine and the epitopes it exposes to the immune system. It also depends on vaccination frequency and the period between vaccine doses which determine the average genetic distance between the vaccine vector and the currently circulating viral variant. The more recent the vaccine vector is the stronger selection pressure to escape immune response in vaccinated individuals it exerts on the virus. This relationship is determined by an important immunological parameter, termed "cross-immunity function" that represents the decrease in the host susceptibility with the genetic distance $x$ between the infecting virus and a virus from which the individual recovered in the past[68,73–75]. The form of the cross-immunity function determines the cross-recognition of epitope variants and may vary among epitopes, host organisms, and viruses. Reconstructing this function in each case remains a challenge but one can estimate from viral genome sequence data the cross-reactivity half-distance, i.e., the genetic distance where the host susceptibility is half of that in a naïve individual. For example, for influenza A H3N2 the cross-reactivity half-distance of 15 amino acid substitutions has been inferred for humans[73] and measured for equines[109]. For SARS-CoV-2, these estimates are still to be determined.

To get an idea about the magnitude of the effect of vaccination on the substitution rate in epitopes, let us consider a simple example. We assume (i) a vector vaccine similar to Sputnik V or that generated by AstraZeneca that has the entire spike protein

with the same epitopes as the natural virus; (ii) vaccine vector is based on a virus variant similar to a variant that infected a typical individual recovered from natural infection; (iii) immunity after natural infection and vaccination are similar. Suppose that 1% of the population are naïve susceptible, 9% of the population had a natural infection and recovered, and the remaining 90% of individuals were vaccinated. With these assumptions, we have $90/9 = 10$-fold more individuals with vaccine-induced than natural immunity, and hence $s_{vac} \sim 10s$. If $\log N_{inf}$ decreases due to vaccination by the factor of 2, as we estimated above, the substitution rate, which is proportional to both $s_{tot}$ and $\log N_{inf}$, increases by the factor of $\sim 1/2 \times 10 = 5$.

If the ratio of the recovered and vaccinated was different from 1:10, the effect on the substitution rate in epitopes would differ from this estimate. The schematic of the dependence of the substitution rate in epitopes for varying proportions of vaccinated and recovered population is shown in Fig. 4. The case of some African countries which had multiple waves of SARS-CoV-2 infection and hardly any vaccination corresponds to a small proportion of the vaccinated population (extreme case: 0%). The case of some European countries with massive vaccination efforts corresponds to a very high proportion of the vaccinated population (extreme cases 100% achieved in, e.g., the Portuguese elderly). Seasonal influenza with its annual vaccination campaigns in many countries with temperate climates would also correspond to a rather small proportion of the vaccinated population, as compared to the global mass vaccination against SARS-CoV-2 realized within 2 years.

We also assumed that the effects of vaccinal and natural immune response on selection pressure are the same. In fact, such symmetry is unlikely, because the number of immune memory cells against an epitope induced by vaccination and natural infection may differ. Figure 4 shows schematically how the substitution rate in epitopes changes with the relative protection (and hence, the selection pressure on epitopes) rendered by the vaccine as compared to the natural infection (compare the red line with the green and blue lines). In addition, vaccines are composed of a section of the spike protein, and the immune system generates antibodies against other viral proteins as well. Thus, the effect of vaccination on the substitution rate in an epitope can be either stronger or weaker than the effect of the natural immune response. Furthermore, the genomic regions where evolution is accelerated will also differ between vaccinal and natural responses. A detailed study based on a mathematical model and immunological data is required to calculate the acceleration of evolution in various epitopes.

For the sake of simplicity, in our example, almost everyone is either recovered or vaccinated, and the overlap between the two groups is neglected. In reality, there are many people who first recovered from natural infection and then got vaccinated, and vice versa. The overlap might lead to a further increase in the speed of evolution, due to the combined effect of natural and induced immunity, however, the interaction between the two is not trivial.

To summarize, we arrived at very different conclusions regarding the effect of vaccination on the speed of antigenic evolution compared to the previous work[78,80,81] by exploiting the similarities in the molecular structure of antibody-binding regions of different viruses and using the modern theory of multi-locus evolution driven by the immune response. The above example illustrates that, despite the reduction of viral transmission by vaccination, viral evolution in neutralizing antibody epitopes may be accelerated by vaccination several-fold. The transient decrease in the number of infections is outweighed by a stronger immune pressure to change. The cost of the transient reduction in virus circulation is the emergence of more transmissible escape mutants and, hence, a higher number of infected individuals in the population in the future.

## Future research
If SARS-CoV-2 continues to cause the substantial burden of severe disease in vulnerable individuals, we should either design a type of vaccine that does not carry any potential danger of accelerating virus evolution in epitopes but is still effective against severe disease, or find other methods of reducing virus circulation. Several research lines could be pursued further to investigate these options.

**Evolution of epitopes**. The vaccine design that prevents all virus evolution in antibody-binding regions is not obvious. If we use a vaccine based on old strains, with a large genetic distance to the current strain to keep selection pressure at bay, it would not protect against the virus well. If a vaccine is based on recent strains, it would protect well but it would create a strong selection pressure to change epitopes. We could gain an initial understanding of how to develop an optimal vaccination strategy using mathematical models that combine SARS-CoV-2 epidemiology and evolution[64,69–75]. Models of this type could predict the substitution rate in epitopes as a function of relevant vaccination parameters, and should be extended to include age-dependent transmission, vaccination coverage, vaccine efficacies against different outcomes (infection, severe disease, hospitalization, and death), and the genetic distance of the vaccine vector to the current variant. Using an age-structured model, we can find model parameters (e.g., age-specific vaccination coverages) that balance the decrease in infection incidence against the extra selection pressure due to vaccination, for various vaccination objectives. An example of the objective could be optimizing the substitution rate against the number of hospitalizations and deaths in vulnerable individuals in the long term.

The Red Queen process is observed in neutralizing antibody epitopes of SARS-CoV-2[47] that are not functionally critical to the virus and hence can mutate without much decrease in $R_0$ (low mutation cost). Virus evolution due to such low cost of mutations causes a rather rapid decrease in protection after infection or vaccination. This leads to the need for repeated rounds of vaccination potentially further accelerating virus evolution. However, SARS-CoV-2 is controlled also by other important immune mechanisms such as helper CD4 T-cells and cytotoxic CD8 T-cells (CTL)[110]. CTL immune response in a host lowers the viral load and hence the transmission rate at the population level. This effect is incorporated in the reproduction number, $R_0$, which affects the selection pressure of antigenic escape, as given by Eq. 1. The waning antibody protection after vaccination may not necessarily lead to full susceptibility to severe disease because the immune system's memory for T-cells can soften the consequences of repeated infection[18].

The selection pressure at the population level for CTL epitopes to mutate is smaller than for antibody-binding regions. Indeed, their location depends on a highly-variable HLA-subtype and hence varies between individuals. Furthermore, the cost of mutation in CTL epitopes is not as low as for antibody-binding regions in receptor proteins that evolved to mutate easily. The mutation cost of T-cell epitopes is distributed broadly. It varies from very functionally important and hence "expensive" epitopes, where more longevity is expected, to epitopes with a low cost where the virus can mutate because the loss of recognition outweighs the cost of mutation. The mutational trajectory for CTL epitopes and their mutation cost have been studied intensely

in the context of HIV infection[58,111–116]. It has been predicted that CTL epitope mutations start from a cheap mutation with a large loss of recognition moving towards more expensive mutations with a larger cross-reactivity[60]. This accumulated experience may help to develop T-cell vaccines against SARS-CoV-2 in the future.

**Evolution of fitness**. In addition to the evolution in epitopes relevant to immune protection that we have considered before, SARS-CoV-2 evolves almost everywhere in the genome producing new VOCs with increased transmissibility and changed virulence. For example, Omicron BA.1 and BA.2 were more transmissible and less virulent, while there was evidence for increased virulence for Alpha and Delta VOCs[8]. In order to predict the evolution of SARS-CoV-2 properly, we need to know which positions in the viral genome are easier and which are harder to mutate. In other words, we must know the fitness landscape including the fitness effects of individual mutations, the interactions between them and, ultimately, the fitness of entire viral variants. The fitness landscape can be determined directly from viral genome sequence data using a toolbox of new methods developed within the last decade. The fitness of viral variants can be estimated by phylogenetic methods[67,117] with the use of machine learning. The fitness effects of mutations and their interactions can be found from the short-term extrapolation of variant dynamics in time[71] or from rather popular statistical methods based on the so-called "quasi-linkage equilibrium" approximation that neglects genetic linkage between evolving residues[118]. The method was applied to related species for proteins of various organisms[119–123] and, recently, to SARS-CoV-2 using different species of coronavirus[124]. A recent method developed for the multi-locus regime that does not neglect genetic linkage and hence can be applied to genome sequence data from either different species or to the same species, also permits to predict fitness effects of mutations and their interactions[125,126]. The availability of these and related methods makes us confident that soon we will have accurate and regularly updated fitness maps of SARS-CoV-2, enabling the prediction of its potential genetic trajectories to the extent made possible by current methods.

**Evolution of virulence**. Devising future vaccination strategies will require predicting the evolution of virulence. This topic remains a major concern and must be a subject of additional research. Although viruses sometimes decrease virulence to adapt to a host population like seasonal coronaviruses, this is not the general rule. Various models suggest that the evolution towards the increase of virulence does not need to be monotonous in the sense of continuing virulence decrease[127–129]. Some modeling studies demonstrate that the selection for low virulence quantified by mortality is rather weak[19,129]. Indeed, severe outcomes or death occur late into infection, after the person has already infected most of the potential contacts and has been isolated. The general evolution of virulence is determined, primarily, as the evolution towards maximizing the reproduction number, controlled by the trade-off between two factors[127,128]. One factor is the increase in viral fitness due to increased infectivity and transmitted dose. The other factor is the decrease in the period where infected individuals infect others, which is limited by the death or the onset of severe symptoms.

The direction of the evolution of SARS-CoV-2 virulence is rather complex, difficult to predict, and has been a topic of debate since the beginning of the pandemic[128–130]. More virulent variants of SARS-CoV-2 have evolved and could still arise in the future if they have higher transmissibility[19]. The Alpha variant was more virulent than the original variant[131], Delta was more virulent than Alpha, but Omicron BA.1/BA.2 was much less virulent than Delta. There are also numerous examples where evolution within a host is the cause of virulence. For example, the evolution of polio is the reason for brain damage[132], and HIV evolution is a possible reason for the onset of AIDS symptoms[133]. HIV does not decrease its virulence in time and kills almost all infected individuals if left untreated. A highly virulent variant of HIV-1 was recently found in the Netherlands[134]. It is also not clear whether SARS-CoV-2 could evolve towards higher virulence in younger age groups[17,22]. If this possibility is ever realized, it could have major consequences for COVID-19 control.

**Origin of VOCs**. Another puzzle important to solve for devising future vaccination strategies for vulnerable individuals is the origin of VOCs that have dozens of new mutations at once, with a temporary acceleration of the evolution rate[8,135]. Alternative theories of the emergence of VOCs[19] include reverse zoonosis, evolution within immunocompromised patients with chronic infection[136–138], and evolution in subpopulations not covered by genetic surveillance. To all these diverse hypotheses, we can add the fitness valley effect, a cascade emergence of compensating mutations following a primary mutation inferred for HIV and influenza[126,139]. Primary mutations in HIV are caused by the early immune response in CTL epitopes. Because these primary mutations decrease the ability of the virus to replicate, mutations on sites located outside of epitopes partly compensating for this decrease come under positive selection. Alleles with a stronger epistatic interaction with the primary sites sweep first[139]. About half of the epitopes do not undergo antigenic escape and are left to limit virus replication[58,140]. The effect could be investigated using methods developed to measure interactions between mutations at different genomic locations[119,123,124,126,141,142].

The emergence of VOCs could also have to do with the effects of natural selection and recombination. The latter occurs when two or more viral variants fuse their genetic material into a new variant during a co-infection in a host. Natural selection and recombination could group and amplify a broadly variable number of beneficial mutations. A deeper insight into SARS-CoV-2 recombination[143–147] is therefore needed to understand its potential implications for future vaccination plans. To quantify recombination, we need to estimate the effective outcrossing number between different SARS-CoV-2 variants that is currently unknown. This number, in principle, can be inferred from genomic samples by fitting the results of simulated models. Methods previously developed to quantify recombination using genome sequence data for HIV could be applied to SARS-CoV-2 as well[148]. The outcrossing number for SARS-CoV-2 could be quite large due to the possibility of co-infection during super-spreading events[149–152].

**Vaccine design**. From a practical viewpoint, SARS-CoV-2 as a public health problem could be solved by pursuing a better vaccine design. The protection mechanism of current vaccines is based on eliciting in vaccinated individuals the neutralizing antibody response against spike protein that imitates the response after natural infection. However, the virus evolved a specific design of spike similar to anti-receptors of HIV and influenza[153]. The receptor-binding site that serves for cell entrance is hidden from antibodies, and it has a conserved sequence. In contrast, the protruding regions of the spike exposed to antibodies are not functionally important and can easily mutate to decrease antibody binding.

The question is then how to achieve vaccine designs that increase the cost of escape mutations. Attempts to produce

synthetic vaccines from chosen sets of conserved virus epitopes prove difficult due to the protein folding of the constructs that hide the intended epitopes from antibodies. Another direction could be therapeutic antibodies or next-generation vaccines based on broadly neutralizing antibodies that target conserved regions of the spike protein and thus make escape more difficult[154]. The immune system mounts the T-cell response against SARS-CoV-2[155]. The development of T-cell-based vaccines is also interesting because some epitopes may be too expensive to mutate. This effect was observed before in HIV[58,111–116] and could be relevant for SARS-CoV-2 as well.

**Defective interfering particles and other options**. Live vaccines could be another option. Their modern formulations could be based on the concept of defective interfering particles (DIPs). DIPs are parasites of parasites, i.e., viruses that cannot exist without the main virus because they lack critically important proteins. In addition to suppressing the virus by stealing its proteins[156–159], a DIP can also serve as a live vaccine eliciting an immune response[160]. Prototypes of DIPs demonstrating both in-host viral suppression and vaccination properties have been recently developed for SARS-CoV-2[156,160]. Importantly, a DIP is under pressure to co-evolve with the main virus and may be actually co-stable[157,158]. Next-generation medicines such as oral antiviral agents and monoclonal antibody prophylaxis will also be needed, especially for immunocompromised people for whom vaccination is not effective.

## Conclusion

Mass vaccination has been very effective in reducing deaths, severe disease, and overall disease burden due to COVID-19 in many countries. At the same time, SARS-CoV-2 has evolved to escape to some extent from both natural and vaccine-induced immunity. From this Perspective, we discussed the possibility that global vaccination may accelerate SARS-CoV-2 evolution in rapidly mutating antibody-binding regions compared to natural infection. Our conclusions rely on the assumption that immune selection pressure acting on the antibody-binding regions of SARS-CoV-2 is similar to that of influenza, and on existing multi-locus models of influenza evolution. To this end, we took advantage of the similarity between the envelope proteins of the two viruses and the evidence of strong genetic variation in the antibody epitopes. The validity of our assumption remains to be tested directly for SARS-CoV-2 in the future, as it was done for the influenza virus. The potential impact of vaccination on SARS-CoV-2 evolution should be acknowledged for future vaccination strategies that target most at-risk populations, especially if vaccination campaigns will cover a substantial part of the population. Mutations in immunologically-relevant genomic regions, viral recombination, virulence, and fitness evolution must be considered when designing a future vaccination strategy. Finally, we would like to stress that despite the potential implications of vaccination for evolution in the antibody epitopes, in the face of an unprecedented global health crisis like the one we just experienced, mass vaccination is probably the only tool to prevent widespread loss of human lives and huge economic costs.

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

## Acknowledgements

We thank Jantien Backer (The National Institute for Public Health and the Environment, The Netherlands), Patricia Bruijning-Verhagen (University Medical Center Utrecht, The Netherlands), Mirjam Kretzschmar (University Medical Center Utrecht, The Netherlands), Peter Lidsky (University of California San Francisco, USA), Richard Neher (University of Basel, Switzerland), Anna Nunes (Lisbon University, Portugal), Michiel van Boven (The National Institute for Public Health and the Environment, The Netherlands), Christiaan van Dorp (Columbia University, USA), and Andrey Vasin (Peter the Great St. Petersburg Polytechnic University, Russia) for useful comments. G.R. was supported by the VERDI project (101045989), funded by the European Union. Views and opinions expressed are however those of the author(s) only and do not necessarily reflect those of the European Union or the Health and Digital Executive Agency. Neither the European Union nor the granting authority can be held responsible for them. G.R. was supported by the Fundação para a Ciência e a Tecnologia project 2022.01448.PTDC. G.R. was supported by the ZonMw project number 10430362220002. I.M.R. worked within the framework of the state assignment of the Federal Agency for Scientific Organizations (FASO Russia: topic no. AAAA-A18-118012290142-9).

## Author contributions

Conception and design and drafting of sections: I.M.R. and G.R.

## Competing interests

Ganna Rozhnova is an Editorial Board Member for Communications Medicine, but was not involved in the editorial review of, nor the decision to publish this article. The authors declare no competing interests.
