## [Peer Review File · Communications Medicine]

Reviewers' comments:

Reviewer #1 (Remarks to the Author):

This manuscript is an interesting and generally well-considered perspective on vaccine design and rollout strategies. It is timely in that globally public health professionals are wrestling with such topics around Sars-CoV-2 and are likely to continue doing so over the next years.

Within this manuscript, the authors summarized the epidemiological motivation behind vaccination strategies and the effectiveness of vaccines towards reducing disease and lessening burden on health care systems. Some of the writing is in need of a thorough edit for clarity and precision. The authors' discussion of the evolutionary implications of vaccinations was very intriguing, although at times relied on a series of large assumptions that would benefit from some expansion and clarification:

1) natural infection and vaccinations induce comparable immune protection,

-this assumption seems unlikely to be the case considering for example that vaccines are composed of a section of the Spike protein and the immune system generates antibodies also against other viral proteins as well. The authors should clarify and expand upon this assumption in greater detail.

2) genetic distance is the same as antigenic distance

-this is also a significant assumption that would benefit from some further evidence/discussion/expansion. As a small number of genetic changes may induce large changes in antigenic differences the relationship thus may not be linear.

3) the log relationship between the number of infected individuals, N_{inf} and the speed of evolution, V .

-Further, clarity for a general audience on the details of how this would be expected to affect the speed of evolution would be valuable.

Generally, further detail on these assumptions would be valuable and greatly improve the manuscript perhaps with explanatory figures.

A clearer discussion/contextualization of the speed of evolution of Sars-CoV-2 relative to other viruses would be useful. As written because Sars-CoV-2 evolution is described as "astonishingly fast" alongside HIV the reader is left with the impression that Sars-CoV-2 is a rapidly evolving virus when in fact relative to HIV and HCV it is very slow. In general viruses like measles would be considered slowly evolving, viruses like HIV and HCV would be considered rapidly evolving and Sars-CoV-2 which accretes about 2 substitutions/month would be in between. A figure with time on the x and evolutionary rate on the y highlighting the evolutionary rate of different viruses would clarify this relationship more clearly for the reader.

A further point that, the authors could profitably consider more carefully is that the “evolution of virulence” section is currently framed with the idea that viruses generally evolve to become less virulent (line 453-455: “According to conventional wisdom and many observations, virus gradually evolves its virulence down in order to enhance it’s transmission, and we have already observed this in the case of Omicron.” This is incorrect please see this recent paper (among many others) which addresses this topic: <https://www.science.org/doi/10.1126/science.abm4915>

A relevant excerpt pasted below from Koelle et al. below:

“The evolution of SARS-CoV-2 virulence in terms of how harmful or deadly it is has been a topic of debate since the beginning of the pandemic (83). Evolutionary theory has pointed out that we should not expect evolution toward lower virulence (84), and the last two variants of concern have demonstrated that there is not a clear, consistent trend in SARS-CoV-2 virulence evolution: Although Delta is thought to be slightly more virulent than previous variants, Omicron is less so. Although the virulence of SARS-CoV-2 may still evolve over time (in a direction that is not easily predicted), we expect the infection fatality ratio to decline for other reasons, including rising population immunity.”

A more balanced perspective/review of the literature in the area of virulence evolution with respect to Sars-CoV-2 immunity would be highly relevant and advisable here. Some other examples are provided e.g. polio however the reader is left with the incorrect impression that in general evolution favours lowering of virulence over time when in fact the tradeoffs that control the evolution of viral virulence are highly complex and difficult to predict.

Other Suggestions

L26: replace “evolution theory” with “evolutionary theory”

L32-33: The authors’ assertion that COVID has required ‘the largest control effort since the 1918 influenza epidemic’ is not necessarily an objective truth, as HIV and smallpox eradication, for example, have required massive public health efforts.

L33: Should be ‘Wuhan, China’

L34: Although VOCs were not detected until late 2020, many of the first samples were collected earlier in 2020 (see outbreak.info) suggesting they emerged within the first year of the pandemic.

L45: Cannot assert that VOC emergence coincided with vaccine rollout without estimating their emergence and considering that their first samples (identified retrospectively) were prior to vaccine availability.

L53: “endgame” – suggest changing wording.

L63: “constantly evolves by natural infection” – do you mean natural selection and drift?

L70: “Since the protective effect of current vaccines is mostly driven by neutralizing antibodies” – citation to support? See <https://www.nature.com/articles/s41590-021-01122-w> on T-cell mediated immunity. I do see that you commented on this in the Discussion, L417.

L73: ‘Epidemiological determinants’ does not seem appropriate to describe all subtitles below.

L83: “The respiratory” should be “Respiratory”

L100: Are children less susceptible to infection (as you insinuate) or less likely to be sampled or have serious outcomes?

Box 1: Suggest ordering the subtitles similarly to text, starting with age. Also, ‘Epidemiological determinants of vaccination’ should be ‘Vaccination strategy considerations’ maybe.

L115: Epidemiological effects, or do you mean sociodemographic determinants?

L124: Should be ‘co-morbidities’.

L148: ‘allow to evaluate’ – grammar. Better: allow us, or allow the evaluation of.

L180: ‘constantly acquiring new mutations’ – some would argue it is not constant.

L184: ‘motives’ should be ‘motifs’. Also, evolution in spike does not necessarily or deterministically result in VOC with enhanced transmissibility.

L190: epitopes ‘serve primarily as highly-variable decoys for antibodies’ – one could argue this is not their primary function.

L191-192: Awkward wording: “Vaccination meant that these variants spread faster, but it was probably not involved as selection pressure in their genesis”. Also, citation for assertion that vaccinations resulted in faster spread of VOCs? This is speculative. Some VOCs were highly transmissible with no immune evasion characteristics.

L197: Awkward grammar: “can accumulate within one individual as well of the evolution in the rest of SARS-CoV-2 genome and the origin of VOCs.”

L200: Should be ‘allows us to’ or ‘facilitates a decrease in’

L209: Used perpetual and perpetually in back-to-back sentences. Reword.

L214: Suggest changing ‘evolution speed’ to ‘evolutionary rate’ throughout the manuscript.

L227: Other factors affect R_0 and R_e besides viral fitness. Should elaborate on these also. What you are describing with ‘reproductive success of a virus in producing infected progeny’ is burst size or replication competence, not transmissibility.

L232: “ R_0 is much larger than 1” – would be better to provide a range with a citations.

L.237: word choice questionable: ‘permanent’

L267: “genome of SARS-CoV-2 evolves at hundreds of positions at a time” implies that they accrue mutations simultaneously. Rerword to something like ‘sc2 genome has hundreds of evolving sites’ or ‘varying sites’

L313: HIV ART might not stop within-host evolution in reservoirs that are not penetrated by ART, such as the central nervous system. ‘Decelerates’ or ‘impedes’ might be a better word choice.

L359: Selection pressure may also depend on interdose period between vaccines. Opportunities for immune evasion will be greatest at low/intermediate levels of selection pressure after one dose, therefore the period before the second (or subsequent) dose is given will affect the opportunity window for ideal vaccine-induced immune escape.

L377: Should you stratify vaccinated individuals with and without natural infection? Those who are vaccinated and have had natural infections may have both broader and stronger immune responses than those only vaccinated or only naturally infected.

L380: Might be nice to include a figure to demonstrate this point.

L389-91: Could comment on the reduction in virulence observed with recent VOCs that have evolved to evade vaccine-induced immunity, ie Vaccines make the Red Queen run faster, causing Alice to run faster, but the consequence of her not keeping up might not be as grave. On this point, might be worth elaborating on the allegory above, clarifying if Alice is the virus keeping up with the immune system (Queen), or vice versa. Is one chasing the other, or are they chasing each other?

L393: ‘the future’

L397-411: I suggest adding to discussion on targeting conserved versus variable regions of spike, inducing different classes of antibodies. See papers on inducing class 3 and 4 antibodies, for example: <https://www.frontiersin.org/articles/10.3389/fimmu.2021.752003/full>. I also expected to see a mention of the ‘first antigenic sin’, whereby an immune response generated to the earlier Wuhan-hu-1 strain (induced by vaccine or wildtype infection) could reduce protection against divergent strains by biasing the response towards the early wildtype virus.

L433: Casual language: ‘pretty much’

L434, L455: Omicron BA.4 and BA.5 may not be as low of virulence as BA.1, BA.2, BA.3, therefore they should be distinguished.

L470: Replace ‘population pockets’ with ‘subpopulations’

L471: Elaborate on fitness valley effect.

Reviewer #2 (Remarks to the Author):

The authors write a perspective on an important topic to discuss - the degree to which vaccination may accelerate mutations and immune escape in SARSCoV2.

Much of the review is dedicated to a preamble about vaccines (5 pages) - and could be reduced. Even after this a lot is explaining basic epidemiological concepts. Which is of course useful. But as a perspective there is then less substance on the actual matter being discussed until later in the manuscript. It could be made much more concise and focused.

As a general point, a lot of the discussion about vaccine induced immunity doesn't appear to take into account the alternative scenario - that immunity will continue to be generated via multiple waves of infection as has happened in many countries in Africa. How is this immunity going to be better or worse than multiple vaccine doses in driving evolution?

I have some specific points:

I.29 (Abstract) "...a new phase where we fully resumed socio-economic activity...". Needs rephrasing to read more clearly. Also, I think the point does not require any mention of moving towards a phase of fully resumed socioeconomic activity as in most countries we are already there.

I. 44-45 '...the onset of vaccination campaigns nearly coincided with the emergence of viral variants...'. 'Nearly' coincided is very vague. I would remove this as it infers a relationship between early VOC and vaccination (which the authors later in the paragraph highlight is not the case). The reality is there was more evidence of positive selective pressure on the virus from around October to November 2020 (Martin et al, Cell, PMID 34537136), which was before any mass vaccination campaigns. Also, these campaigns were either delayed or never had high population coverage in many countries where VOC may have emerged. These were most likely driven by other factors (e.g. alpha), including immunity after infection in the case of variants that had some antibody evasive properties (e.g. beta).

L. 62 '...SARSCoV2 evolves by natural infection but also by vaccination..'. This sentence needs rephrasing. I think the authors mean to say that immunity induced by infection and vaccination both may result in selective pressure to evolve.

L. 61 - 66. This paragraph could be refined as it repeats the point about infection and vaccination both exerting some selective pressure. What isn't mentioned here though is the fact that pressure from most vaccines currently is only on Spike, whereas infection may induce pressure on other areas. So vaccination only increases immune pressure on spike evolution.

I. 112 - properties and performance of vaccines. One key aspect that is missing in this

paragraph is that when it comes to infection blocking or transmission blocking properties, that the effectiveness is time limited and reduces the longer one gets from the last dose. l. 126 - 128 could be improved. 'It is never perfect' is very vague. The influenza vaccine comment also needs referencing and making more specific. At present it reads that 'influenza vaccines are not perfect at protecting against infection'. But how? on what evidence?

Where referring to Omicron (e.g l.191) - refrain from saying 'current VOC' - BA.1 is already not the current VOC

l.191 'Vaccination meant that these variants spread faster' - I think this needs rephrasing. Vaccination and therefore enhanced population immunity has meant that omicron has an advantage over its predecessor variants with less mutations impacting Nab (so higher comparative growth rate). This does not mean vaccination = faster spread of a virus.

l. 201 - 'in the longer term, however, mass vaccination further accelerates the rapid evolution of epitope regions'. This is stated as fact, which is inappropriate. It is a fair hypothesis to have and discuss, but should not be stated in this way prior to any discussion of why this statement might be true.

l. 203 - 211. This paragraph does not mention the fact that influenza vaccination happens every year, at least in many countries with temperate climates

Fig 2 - please define Ub in the legend as it isn't clear just looking at the figure what 'NinfUb' is

l. 320 - 322. Again, the authors state as fact that vaccination accelerates virus evolution in antibody-binding regions. This can be discussed as a hypothesis but needs to be moderated. The key question is does vaccination increase evolution compared to the alternative scenario - which is waves of infection induced immunity, which will also then wane. The current discussion is very focused around the theoretical concepts of vaccine immunity driving evolution - which is possible. But it can't be discussed as if the alternative is 'no immunity' and therefore no selective pressure. I am not sure that there is enough evidence in what the authors present for vaccination 'favouring immune escape mutations even more' (l.330 - 331)

Immunocompromised individuals and the relevance to mutations is mentioned at the end of the manuscript. However, if as is the current hypothesis, most VOC that have led to sweeping replacements globally originated in this way, then it is hard to argue that mass vaccination will play any role. It may be that SARSCoV2 evolution settles into a more 'ladder-like' pattern like H3N2 along Omicron lineages (where it would then be more relevant), but we simply do not know yet.

REVIEWERS' COMMENTS:

Reviewer #1:

Note to the Reviewer: Lines and pages in our response refer to the resubmitted manuscript with marked changes, unless stated otherwise.

Newly added references: 9, 37, 44, 53-55, 71, 72, 87, 88, 89, 98, 120, 122, 126, 145, 146, 152, 153, 154.

Newly added figures: Figure 1, Figure 4.

This manuscript is an interesting and generally well-considered perspective on vaccine design and rollout strategies. It is timely in that globally public health professionals are wrestling with such topics around Sars-CoV-2 and are likely to continue doing so over the next years.

We thank the Reviewer for finding our Perspective interesting, timely and generally well-considered.

Within this manuscript, the authors summarized the epidemiological motivation behind vaccination strategies and the effectiveness of vaccines towards reducing disease and lessening burden on health care systems. Some of the writing is in need of a thorough edit for clarity and precision. The authors' discussion of the evolutionary implications of vaccinations was very intriguing, although at times relied on a series of large assumptions that would benefit from some expansion and clarification.

We fully agree with the Reviewer that some assumptions needed further expansion and clarifications which we have done in the revised MS as follows:

1) natural infection and vaccinations induce comparable immune protection

-this assumption seems unlikely to be the case considering for example that vaccines are composed of a section of the Spike protein and the immune system generates antibodies also against other viral proteins as well. The authors should clarify and expand upon this assumption in greater detail.

We agree with the Reviewer. In the revised MS, we added a new figure (Figure 4) and expanded upon this assumption as follows (Lines 650-660):

"We also assumed that the effects of vaccinal and natural immune response on selection pressure are the same. In fact, such symmetry is unlikely, because the number of immune memory cells against an epitope induced by vaccination and natural infection may differ. Figure 4 shows schematically how the substitution rate in epitopes changes with the relative protection (and hence, the selection pressure on epitopes) rendered by vaccine as compared to the natural infection (compare the red line with the green and blue lines). In addition, vaccines are composed of a section of the spike protein, and the immune system generates antibodies against other viral proteins as well. Thus, the effect of vaccination on the substitution rate in an epitope can be either stronger or weaker than the effect of the natural immune response. Furthermore, the genomic regions where evolution is accelerated will also differ between vaccinal and natural responses. A detailed study based on a mathematical model and immunological data is required to calculate the acceleration of evolution in various epitopes."

2) genetic distance is the same as antigenic distance

-this is also a significant assumption that would benefit from some further evidence/discussion/expansion. As small number of genetic changes may induce large changes in antigenic differences the relationship thus may not be linear.

Thank you for this suggestion. We expanded on this assumption as follows (Lines 513-528):

“The studies ⁶⁸⁻⁷⁰ assume that the genetic distance is linearly proportional to antigenic distance. An additional complication is that different mutations in epitopes have a variable effect on epitope binding. Some mutations make a large change in the free energy of binding, and some mutations are almost neutral. In other words, the genetic distance and antigenic distance are not linearly proportional. In terms of the phenotype landscape, it has a rugged component. This fact has been investigated in detail using two-dimensional antigenic maps for influenza ^{56,65} and is of major importance when the short-term evolution is studied. In the long-term evolution, an effective ratio of antigenic to genetic change can be successfully described by studies with the averaged-out cross-immunity function ^{66,68-70}.”

3) the log relationship between the number of infected individuals, N_{inf} and the speed of evolution, V .

-Further, clarity for a general audience on the details of how this would be expected to affect the speed of evolution would be valuable.

We added the suggested clarification and refer the reader to Figure 3 (Lines 417-434):

“However, if product $N_{inf}U_b$ is much larger than 1, mathematical models predict that mutations occurring at different positions of the viral genome are concurrent in time, and the approximation of independent sweeps does not apply anymore due to strong interference between different mutations ⁷⁶⁻⁸⁵. This linkage interference creates several effects such as interference between emerging clones (Fisher-Müller effect) ^{86,87}, which is equivalent ⁸⁸ to Hill-Robertson effect ⁸⁹, i.e., the decrease of selection effect at one locus due to selection at another locus, as well as various genetic background effects. All of them are directly observed for seasonal influenza and many other viruses ^{52,86,90,91} (Figure 3b). The linkage interference effects slow down virus evolution by orders of magnitude and change the way the substitution rate ⁷⁶⁻⁸⁵, V , and the statistics of phylogenetic trees ^{92,93} depend on the number of infected individuals, N_{inf} , mutation rate, U_b , and recombination rate.

SARS-CoV-2 genome has hundreds of evolving sites ⁹⁴. The virus demonstrates, on average, more than two substitutions per month, or 1.1×10^{-3} substitutions per site per year ⁴⁴, and modest intra-host diversity ^{95,96}. Its evolution can be described by multi-locus models, which have been studied intensely over the last two decades using the methods of statistical physics ^{76-85,92,93}. Their exact predictions for the substitution rate, V , vary depending on the specific evolutionary factors included into consideration. However, all these models demonstrate that V depends weakly on both the number of currently infected individuals, N_{inf} , and mutation rate, U_b . More specifically, V is linearly proportional to $\log N_{inf}$ and increases logarithmically with U_b as well (Figure 3b).”

Generally, further detail on these assumptions would be valuable and greatly improve the manuscript perhaps with explanatory figures.

As requested, we made the explanations listed above. The log relationship between the number of infected individuals and the speed of evolution is demonstrated in Figure 3b (point 3 by the Reviewer). We also added a new figure (Figure 4, page 15) with the schematic of the dependence of the substitution rate on the proportions of vaccinated and recovered individuals when vaccine efficacy in inducing protection against infection is equal, smaller, and larger than the natural immune response

(Assumption 1). We prefer not to add any figure on Assumption 2 tangential to our main message in this Perspective.

A clearer discussion/contextualization of the speed of evolution of Sars-CoV-2 relative to other viruses would be useful. As written because Sars-CoV-2 evolution is described as “astonishingly fast” alongside HIV the reader is left with the impression that Sars-CoV-2 is a rapidly evolving virus when in fact relative to HIV and HCV it is very slow. In general viruses like measles would be considered slowly evolving, viruses like HIV and HCV would be considered rapidly evolving and Sars-CoV-2 which accretes about 2 substitutions/month would be in between. A figure with time on the x and evolutionary rate on the y highlighting the evolutionary rate of different viruses would clarify this relationship more clearly for the reader.

Thank you for this suggestion. To address it, we have added a bar plot (new Figure 1, Lines 284-297) comparing the average substitution rates of different rapidly-evolving viruses: HIV, HCV, influenza, and SARS-CoV-2. We do not see how the axis of time would be relevant here.

A further point that, the authors could profitably consider more carefully is that the “evolution of virulence” section is currently framed with the idea that viruses generally evolve to become less virulent (line 453-455: “According to conventional wisdom and many observations, virus gradually evolves its virulence down in order to enhance it’s transmission, and we have already observed this in the case of Omicron.” This is incorrect please see this recent paper (among many others) which addresses this topic: <https://www.science.org/doi/10.1126/science.abm4915>

A relevant excerpt pasted below from Koelle et al. below:

“The evolution of SARS-CoV-2 virulence in terms of how harmful or deadly it is has been a topic of debate since the beginning of the pandemic (83). Evolutionary theory has pointed out that we should not expect evolution toward lower virulence (84), and the last two variants of concern have demonstrated that there is not a clear, consistent trend in SARS-CoV-2 virulence evolution: Although Delta is thought to be slightly more virulent than previous variants, Omicron is less so. Although the virulence of SARS-CoV-2 may still evolve over time (in a direction that is not easily predicted), we

expect the infection fatality ratio to decline for other reasons, including rising population immunity.”

A more balanced perspective/review of the literature in the area of virulence evolution with respect to Sars-CoV-2 immunity would be highly relevant and advisable here. Some other examples are provided e.g. polio however the reader is left with the incorrect impression that in general evolution favours lowering of virulence over time when in fact the tradeoffs that control the evolution of viral virulence are highly complex and difficult to predict.

Thank you for the valid point and the references. The section has been rewritten, as follows (Lines 744-782):

“Obviously, devising future vaccination strategies will require predicting the evolution of virulence. This topic remains a major concern and must be a subject of additional research. Although viruses sometimes decrease virulence to adapt to a host population, this is not the general rule. Various models suggest that the evolution towards the increase of virulence does not need to be monotonous¹¹⁹⁻¹²¹. Some modeling studies demonstrate that the selection for low virulence quantified by mortality is rather weak^{20,121}. Indeed, severe outcomes or death occur late into infection, after the person has already infected most of the potential contacts and has been isolated. The general evolution of virulence is determined, primarily, as the evolution towards maximizing the reproduction number, controlled by the trade-off between two factors^{119,120}. One factor is the increase in viral fitness due to increase in infectivity and transmitted dose. The other factor is the decrease in the period where infected individuals infect others, which is limited by the death or the onset of severe symptoms.

The direction of evolution of SARS-CoV-2 virulence is rather complex, difficult to predict and has been a topic of debate since the beginning of the pandemic¹²⁰⁻¹²². More virulent variants of SARS-CoV-2 have evolved and could still arise in future if they have higher transmissibility²⁰. Alpha variant was more virulent than the original variant¹²³, Delta was more virulent than Alpha, but Omicron BA.1/BA.2 was much less virulent than Delta. There are also numerous examples where evolution within a host is the cause of virulence. For example, the evolution of polio is the reason for brain damage¹²⁴, and HIV evolution is a possible reason for the onset of AIDS symptoms¹²⁵. HIV does not decrease its virulence in time and kills almost all infected individuals if left untreated. A highly virulent variant of HIV-1 was recently found in the Netherlands¹²⁶. It is also not clear whether SARS-CoV-2 could evolve towards higher virulence in younger age groups^{18,23}. If this possibility is ever realized, it could have major consequences for COVID-19 control.”

Other Suggestions

L26: replace “evolution theory” with “evolutionary theory”

Done (Line 26).

L32-33: The authors’ assertion that COVID has required ‘the largest control effort since the 1918 influenza epidemic’ is not necessarily an objective truth, as HIV and smallpox eradication, for example, have required massive public health efforts.

Changed to (Lines 35-36):

“required massive control efforts like the 1918 influenza pandemic, HIV and smallpox eradication”.

L33: Should be ‘Wuhan, China’

Fixed (Line 37).

L34: Although VOCs were not detected until late 2020, many of the first samples were collected earlier in 2020 (see [outbreak.info](https://www.outbreak.info)) suggesting they emerged within the first year of the pandemic.

Changed to (Lines 36-38):

“SARS-CoV-2 variants that were substantially more transmissible or caused more severe disease than the original variant from Wuhan, China were not detected until late 2020.”

L45: Cannot assert that VOC emergence coincided with vaccine rollout without estimating their emergence and considering that their first samples (identified retrospectively) were prior to vaccine availability.

The sentence was changed to highlight ‘detection’ and not ‘emergence’ (Lines 47-51):

“However, the onset of vaccination campaigns nearly coincided with the detection of viral variants, dubbed “variants of concern” (VOCs) ^{7,8}, that differed from the previous variants due to their demonstrated impact on transmissibility ⁹⁻¹¹, disease severity ¹²⁻¹⁵ and the ability to evade a host’s immune response ¹⁶ after natural infection or vaccination.”

NB! We actually agree. In our original MS we had (Lines 48-49):

“Before the current VOC, Omicron, all VOCs had emerged before the mass rollout of vaccination.”

In the revised MS, we expanded on this as follows (Lines 567-569):

“Before Omicron and its descendants, all VOCs had emerged before the mass rollout of vaccination as inferred from phylogenetic analyses ^{9,98}. Vaccination was probably not involved as selection pressure in their genesis. The same is likely true for Omicron BA.1.”

L53: “endgame” – suggest changing wording.

Changed to (Line 55):

“reach SARS-CoV-2 control globally”.

L63: “constantly evolves by natural infection” – do you mean natural selection and drift?

Changed to (Lines 62-67):

“SARS-CoV-2 perpetually evolves due to its escape from the immune response in individuals induced by both natural infection and vaccination. Even in the absence of vaccination, there is selection pressure to escape natural immunity by accumulating mutations in T-cell epitopes and antibody-binding regions. Mass vaccination, as we show below, increases this pressure and accelerates SARS-CoV-2 evolution in spike epitopes compared to natural infection.”

L70: “Since the protective effect of current vaccines is mostly driven by neutralizing antibodies” – citation to support? See <https://www.nature.com/articles/s41590-021-01122-w> on T-cell mediated immunity. I do see that you commented on this in the Discussion, L417.

Changed to (Lines 92-94):

“The current vaccines are designed to induce mostly neutralizing antibody response against spike protein, therefore we focus on the evolution of rapidly mutating antibody-binding regions.”

We have also added the above reference in the discussion of T-cell response at the end (Lines 836-837):

“The immune system mounts the T-cell response against SARS-CoV-2 ¹⁴⁶. The development of T-cell-based vaccines is also interesting because some epitopes may be too expensive to mutate.”

L73: ‘Epidemiological determinants’ does not seem appropriate to describe all subtitles below.

We changed ‘epidemiological determinants’ to ‘vaccination strategy considerations’ (Line 96).

L83: “The respiratory” should be “Respiratory”

Fixed (Line 106).

L100: Are children less susceptible to infection (as you insinuate) or less likely to be sampled or have serious outcomes?

It is a bit strange to use the term ‘insinuate’ after we supported our statement by references 26, 27, 33, 34, and 35 (now Lines 120-123, 160-161). In certain types of studies, e.g., household studies, all individuals can be sampled independently of any symptoms. We will not expand on this topic further because it is not essential to our manuscript. For clarity, we inserted “susceptible to infection” (Lines 122-123):

“Children and adolescents younger than 20 years are estimated to be about 50% less susceptible to infection than adults who are 20 years and older, and very young children even less ^{27,35}.”

Box 1: Suggest ordering the subtitles similarly to text, starting with age. Also, ‘Epidemiological determinants of vaccination’ should be ‘Vaccination strategy considerations’ maybe.

We changed ‘epidemiological determinants’ to ‘vaccination strategy considerations’ in Box 1 and the subtitle in the text (Line 96, Line 100). We also reordered and renamed some subtitles in the text and in Box 1. The first subtitle in the box is not discussed in the text in detail due to a word limit. This subtitle lists common facts that logically should follow before any discussion on the importance of the age. The box is not intended to repeat the text and serves as a summary relevant for SARS-CoV-2.

L115: Epidemiological effects, or do you mean sociodemographic determinants?

We rephrased this sentence as follows (Lines 175-177):

“Regardless of their formulation, all vaccines approved by major public health agencies were shown to have three main protection effects (reviewed in ³⁶) that are important for devising vaccination strategies (Box 1).”

Then, we explain the effects of vaccines further down.

L124: Should be ‘co-morbidities’.

Fixed (Line 184).

L148: 'allow to evaluate' – grammar. Better: allow us, or allow the evaluation of.

Fixed (Line 230).

L180: 'constantly acquiring new mutations' – some would argue it is not constant.

Changed to '*perpetually*' (Line 279).

L184: 'motives' should be 'motifs'.

Fixed (Line 283).

Also, evolution in spike does not necessarily or deterministically result in VOC with enhanced transmissibility.

Changed to (Lines 283-286):

"Firstly, spike has receptor-binding motifs that affect transmission, and their evolution leads to the increase in virus fitness. This may play a role in the emergence of VOCs with enhanced transmissibility^{10-12,49}."

L190: epitopes 'serve primarily as highly-variable decoys for antibodies' – one could argue this is not their primary function.

We added the explanation as follows (Lines 292-300):

"Thirdly, these epitope regions have evolved to have low physiological constraints on mutation (low mutation cost) because they serve primarily as highly-variable decoys for antibodies. Had they another important function for a virus, they would be conserved. For example, the receptor binding site has a function and is conserved, because it hides between the protruding variable regions to prevent antibody binding. This is the case for both influenza virus receptor (hemagglutinin) and HIV receptor (gp120)⁵²⁻⁵⁵."

L191-192: Awkward wording: "Vaccination meant that these variants spread faster, but it was probably not involved as selection pressure in their genesis". Also, citation for assertion that vaccinations resulted in faster spread of VOCs? This is speculative. Some VOCs were highly transmissible with no immune evasion characteristics.

Changed to (Lines 568-569):

"Vaccination was probably not involved as selection pressure in their genesis."

L197: Awkward grammar: "can accumulate within one individual as well of the evolution in the rest of SARS-CoV-2 genome and the origin of VOCs."

Changed to (Lines 302-304):

"We postpone until the end the discussion of rare chronic infections, where escape mutations can accumulate within one individual, and of the evolution outside of epitopes."

L200: Should be 'allows us to' or 'facilitates a decrease in'

Fixed (Lines 306-307).

L209: Used perpetual and perpetually in back-to-back sentences. Reword.

Changed to "*continue mutating*" (Line 344).

L214: Suggest changing 'evolution speed' to 'evolutionary rate' throughout the manuscript.

We replaced 'evolution speed' with the standard term in population genetics, "substitution rate". It is defined for the first time as (Lines 404-405):

"... the speed of antigenic evolution or substitution rate, V , defined as the rate of accumulation of non-synonymous mutations in all neutralizing antibody epitopes."

L227: Other factors affect R_0 and R_e besides viral fitness. Should elaborate on these also. What you are describing with 'reproductive success of a virus in producing infected progeny' is burst size or replication competence, not transmissibility.

To avoid further confusion, we changed this passage to (Lines 354-363):

"The consequences of this process become clear if we introduce two quantities describing the potential of a virus to spread in a population, the basic and effective reproduction numbers, R_0 and R_e (Box 2). Both measure the Darwinian viral fitness on the population level defined as the number of individuals infected by one individual. They incorporate several host-level factors, including the burst size, infectivity, transmission doze, the immune response in a host, and transmission bottlenecks."

L232: " R_0 is much larger than 1" – would be better to provide a range with a citations.

We added a range and a citation as follows (Lines 367-368):

" R_0 is larger than 1 for the original SARS-CoV-2 variant (range 1.9 – 6.5)⁷¹ and for Alpha, Delta, Gamma, Omicron BA.1/2 VOCs^{8,10,12,72}."

L.237: word choice questionable: 'permanent'

We changed this phrase to (Lines 391-393):

"However, as a result of the ongoing evolutionary escape from the immune response after natural infection, as well due to mutations outside of the epitopes including the receptor binding domain, this outcome is altered to the stationary process with seasonal oscillations."

L267: "genome of SARS-CoV-2 evolves at hundreds of positions at a time" implies that they accrue mutations simultaneously. Reword to something like 'sc2 genome has hundreds of evolving sites' or 'varying sites'

Changed to (Line 427):

"SARS-CoV-2 genome has hundreds of evolving sites⁹⁴."

L313: HIV ART might not stop within-host evolution in reservoirs that are not penetrated by ART, such as the central nervous system. ‘Decelerates’ or ‘impedes’ might be a better word choice.

Changed to (Lines 537-539):

“The therapy not only suppresses the number of infected cells by several orders magnitude, but also strongly impedes within-host evolution, because it reduces the initial reproduction number below 1 very rapidly.”

L359: Selection pressure may also depend on interdose period between vaccines. Opportunities for immune evasion will be greatest at low/intermediate levels of selection pressure after one dose, therefore the period before the second (or subsequent) dose is given will affect the opportunity window for ideal vaccine-induced immune escape.

Changed to (Lines 558-560):

“The extent to which vaccination accelerates evolution in antibody-binding regions depends on vaccination frequency, inter-dose period, molecular design of vaccines, and details of immunological and evolutionary dynamics.”

L377: Should you stratify vaccinated individuals with and without natural infection etc?

A comment is added (Lines 661-665):

“For the sake of simplicity, in our example, almost everyone is either recovered or vaccinated, and the overlap between the two groups is neglected. In reality, there are many people who first recovered from natural infection and then got vaccinated, and vice versa. The overlap will lead to a further increase in the speed of evolution, due to the combined effect of natural and induced immunity, however, the interaction between the two is not trivial.”

L380: Might be nice to include a figure to demonstrate this point

We added a new figure, Figure 4 (page 15), to illustrate the dependence on the proportion of the vaccinated and the naturally-infected previously, and on the relative efficacy of vaccine compared to the natural immune response. We assume that almost everyone is either vaccinated or recovered (see above).

L389-91: Could comment on the reduction in virulence observed with recent VOCs that have evolved to evade vaccine-induced immunity, ie Vaccines make the Red Queen run faster, causing Alice to run faster, but the consequence of her not keeping up might not be as grave. On this point, might be worth elaborating on the allegory above, clarifying if Alice is the virus keeping up with the immune system (Queen), or vice versa. Is one chasing the other, or are they chasing each other?

As we highlighted in the revised MS on the request of the Reviewer, the virulence evolution is complex and difficult to predict (Lines 744-782). We also see that our analogy with the Carroll's characters might be misunderstood, because "Red Queen" is just the name of an effect in population genetics based on the famous phrase. There is no exact allegory with the characters representing virus, immune system etc. To reduce the opportunity for such over-interpretation of the book by Lewis Carroll, we deleted the two sentences from the revised manuscript:

"Vaccines make the Red Queen run faster. Apparently, this is what we are observing at the moment."

L393: 'the future'

Fixed (Line 672).

L397-411: I suggest adding to discussion on targeting conserved versus variable regions of spike, inducing different classes of antibodies. See papers on inducing class 3 and 4 antibodies, for example: <https://www.frontiersin.org/articles/10.3389/fimmu.2021.752003/full>. I also expected to

see a mention of the ‘first antigenic sin’, whereby an immune response generated to the earlier Wuhan-hu-1 strain (induced by vaccine or wildtype infection) could reduce protection against divergent strains by biasing the response towards the early wildtype virus.

We address this briefly in section “Vaccine design” starting on Line 824. In particular, we added the reference suggested by the Reviewer and the following sentence (Lines 833-836):

“Other direction could be therapeutic antibodies or next generation vaccines based on broadly neutralizing antibodies such as class 3 and class 4 antibodies that target conserved regions of spike and thus make escape more difficult¹⁴⁵.”

We have not added a comment on the antigenic imprinting for SARS-CoV-2 that is a current area of research with still limited knowledge. For example, a recent study shows that imprinting differs between natural infection and vaccination:

[https://www.cell.com/trends/immunology/fulltext/S1471-4906\(22\)00048-5](https://www.cell.com/trends/immunology/fulltext/S1471-4906(22)00048-5)

L433: Casual language: ‘pretty much’

Fixed (Line 724).

L434, L455: Omicron BA.4 and BA.5 may not be as low of virulence as BA.1, BA.2, BA.3, therefore they should be distinguished.

We updated the text by changing the sentence to (Lines 725-727):

“For example, Omicron BA.1 and BA.2 were more transmissible and less virulent, while, at the time of writing, no evidence exists for reduced virulence for Omicron BA.3, BA.4 and BA.5⁸.”

L470: Replace ‘population pockets’ with ‘subpopulations’

Done (Line 787).

L471: Elaborate on fitness valley effect.

Added (Lines 789-794):

“Primary mutations in HIV are caused by the early immune response in CTL epitopes. Because these primary mutations decrease the virus ability to replicate, mutations on epistatically-connected sites located outside of epitopes partly compensating for this decrease come under positive selection. Alleles with a stronger epistatic interaction with the primary sites sweep first¹³¹. About a half of epitopes do not undergo antigenic escape and are left to limit virus replication.”

Reviewer #2:

Note to the Reviewer: Lines and pages in our response refer to the resubmitted manuscript with marked changes, unless stated otherwise.

Newly added references: 9, 37, 44, 53-55, 71, 72, 87, 88, 89, 98, 120, 122, 126, 145, 146, 152, 153, 154.

Newly added figures: Figure 1, Figure 4.

The authors write a perspective on an important topic to discuss - the degree to which vaccination may accelerate mutations and immune escape in SARSCoV2.

We thank the Reviewer for appreciating the importance of our topic. We would like to stress that the focus of our Perspective is on the impact of vaccination on evolution *in rapidly mutating antibody-binding regions* at the population-level, which is stated in the abstract and everywhere throughout the text.

Much of the review is dedicated to a preamble about vaccines (5 pages) - and could be reduced. Even after this a lot is explaining basic epidemiological concepts. Which is of course useful. But as a perspective there is then less substance on the actual matter being discussed until later in the manuscript. It could be made much more concise and focused.

We assume that the Reviewer refers to text in Section “Epidemiological determinants of vaccination” (renamed), which is 3 pages long in the original MS. The Perspective must be written for researchers with different backgrounds including those who are not very familiar with epidemiology. Therefore, we prefer not to shorten this section. Instead, we expanded Sections “Evolutionary consequences of vaccination at the population level” and “Future research”.

As a general point, a lot of the discussion about vaccine induced immunity doesn't appear to take into account the alternative scenario - that immunity will continue to be generated via multiple waves of infection as has happened in many countries in Africa.

There should be some confusion. We respectfully point out that we already discussed antigenic drift due to natural infection and immune response in sections: “Virus evolution in epitopes in the absence of vaccination: The Red Queen effect” and “Models of virus evolution in epitopes at the population level” (Lines 310-528). The vaccine effects are discussed in the next two sections.

How is this immunity going to be better or worse than multiple vaccine doses in driving evolution?

To address this important comment, we included Figure 4 (page 15) that shows schematically how the substitution rate in epitopes changes with the proportions of the vaccinated and recovered population, as well as with the relative protection (and hence, the selection pressure on epitopes) rendered by vaccine as compared to the natural infection. Therefore, the two cases (vaccinated and non-vaccinated) are not binary “alternatives”, but different limits with respect to the proportion of vaccinated population. The case of some African countries corresponds to a small proportion of vaccinated population, and the case of some European countries corresponds to a large proportion of vaccinated.

Newly added Figure (page 15) and text reads (Lines 628-649):

Figure 4. Schematics of the dependence of the substitution rate on the proportions of vaccinated and recovered population and on the relative efficacy of vaccine compared to the natural immune response. The substitution rate in epitopes is shown in the presence (three higher lines) and in the absence (three lower lines) of recovered individuals as the proportion of vaccinated population increases. Red, green and blue lines correspond to vaccine efficacy in inducing protection against infection equal, lower, and higher than the natural immune response. We assume the absence of fully susceptible (naïve) individuals.

“If the ratio of the recovered and vaccinated was different from 1:10, the effect on the substitution rate in epitopes would differ from this estimate. The schematic of the dependence of the substitution rate in epitopes for varying proportions of vaccinated and recovered population is shown in Figure 4. The case of some African countries which had multiple waves of SARS-CoV-2 infection and hardly any vaccination corresponds to a small proportion of vaccinated population (extreme case: 0%). The case of some European countries with massive vaccination efforts corresponds to a very high proportion of vaccinated population (extreme case: 100%, achieved in e.g. the Portuguese elderly). Seasonal influenza with its annual vaccination campaigns in many countries with temperate climates would also correspond to a rather small proportion of vaccinated population, as compared to the global mass vaccination against SARS-CoV-2 realized in the timespan of less than two years.”

I have some specific points:

I.29 (Abstract) “...a new phase where we fully resumed socio-economic activity...”. Needs rephrasing to read more clearly. Also, I think the point does not require any mention of moving towards a phase of fully resumed socioeconomic activity as in most countries we are already there.

We updated the text by removing “where we fully resumed socio-economic activity” from this sentence (Lines 28-30):

“The evolutionary consequences of vaccination should be acknowledged as we move towards a new phase where vaccination strategies might need to be updated.”

I. 44-45 ‘...the onset of vaccination campaigns nearly coincided with the emergence of viral variants...’. ‘Nearly’ coincided is very vague. I would remove this as it infers a relationship between early VOC and vaccination (which the authors later in the paragraph highlight is not the case).

To avoid ambiguity, we changed “emergence” to “detection” of VOC in this sentence (Lines 47-51):

“However, the onset of vaccination campaigns nearly coincided with the detection of viral variants, dubbed “variants of concern” (VOCs) ^{7,8}, that differed from the previous variants due to their demonstrated impact on transmissibility ⁹⁻¹¹, disease severity ¹²⁻¹⁵ and the ability to evade a host’s immune response ¹⁶ after natural infection or vaccination.”

The reality is there was more evidence of positive selective pressure on the virus from around October to November 2020 (Martin et al, Cell, PMID 34537136), which was before any mass vaccination campaigns. Also, these campaigns were either delayed or never had high population coverage in many countries where VOC may have emerged. These were most likely driven by other factors (e.g. alpha), including immunity after infection in the case of variants that had some antibody evasive properties (e.g. beta).

We agree and do mention in the manuscript that most VOCs emerged before the mass vaccination as follows (Lines 567-569):

“Before Omicron and its descendants, all VOCs had emerged before the mass rollout of vaccination as inferred from phylogenetic analyses ^{9,98}. Vaccination was probably not involved as selection pressure in their genesis. The same is likely true for Omicron BA.1.”

The reference suggested by the Reviewer has been added as Ref. 98.

L. 62 ‘...SARSCoV2 evolves by natural infection but also by vaccination..’. This sentence needs rephrasing. I think the authors mean to say that immunity induced by infection and vaccination both may result in selective pressure to evolve.

L. 61 - 66. This paragraph could be refined as it repeats the point about infection and vaccination both exerting some selective pressure. What isn’t mentioned here though is the fact that pressure from most vaccines currently is only on Spike, whereas infection may induce pressure on other areas. So vaccination only increases immune pressure on spike evolution.

The passage on Lines 61-66 including Line 62 in the original MS has been edited as follows (Lines 61-67):

“However, the pillar of the public health response to COVID-19, mass vaccination has also consequences for SARS-CoV-2 evolution in antibody-binding regions located in spike protein. SARS-CoV-2 perpetually evolves due to its escape from the immune response in individuals induced by both natural infection and vaccination. Even in the absence of vaccination, there is selection pressure to escape natural immunity by accumulating mutations in T-cell epitopes and antibody-binding regions. Mass vaccination, as we show below, increases this pressure and accelerates SARS-CoV-2 evolution in spike epitopes compared to natural infection.”

We also explain this point further in the text and in new Figure 4 (page 15) in more detail (Lines 560-660):

“We also assumed that the effects of vaccinal and natural immune response on selection pressure are the same. In fact, such symmetry is unlikely, because the number of immune memory cells against an epitope induced by vaccination and natural infection may differ. Figure 4 shows schematically how the substitution rate in epitopes changes with the relative protection (and hence, the selection pressure on epitopes) rendered by vaccine as compared to the natural infection (compare the red line with the green and blue lines). In addition, vaccines are composed of a section of the spike protein, and the immune system generates antibodies against other viral proteins as well. Thus, the effect of vaccination on the substitution rate in an epitope can be either stronger or weaker than the effect of

the natural immune response. Furthermore, the genomic regions where evolution is accelerated will also differ between vaccinal and natural responses. A detailed study based on a mathematical model and immunological data is required to calculate the acceleration of evolution in various epitopes.”

I. 112 - properties and performance of vaccines. One key aspect that is missing in this paragraph is that when it comes to infection blocking or transmission blocking properties, that the effectiveness is time limited and reduces the longer one gets from the last dose.

In the original MS, we addressed this in the section where we explain why the original vaccination strategy had to be adapted (Lines 168-178). In the current MS, as suggested by the Reviewer, we moved some of this text to the section “Properties and performance of vaccines” (Lines 191-194):

“The vaccine efficacies or real-world effectiveness against the VOCs generally declined when compared with the Wuhan variant ³⁶. The biggest reduction was in the efficacy against infection while the efficacies against hospitalization and death were reduced much less.”

The remaining text is left in the section “Global vaccination rollout” (Lines 261-268):

“It is important to note that, because vaccines were rolled out during the ongoing pandemic, the vaccination strategy had to be adapted over time in response to new VOCs and declining protection after natural infection and vaccination. Other factors that started to play a role in the assessment of vaccination strategies are the duration of immune protection after natural infection or vaccination, as well as the cross-protection from prior infection with one variant against another or other seasonal human coronaviruses ¹⁷. Booster vaccination is seen as a way to increase the protection of vaccinated individuals against the new VOCs and keep COVID-19 hospitalizations and deaths at bay.”

I. 126 - 128 could be improved. ‘It is never perfect’ is very vague.

For our purpose in this Perspective, it is important that vaccine efficacy against infection is not 100%, i.e. not perfect. We clarified this as follows (Lines 185-189):

“It is important to stress that, while SARS-CoV-2 vaccines offer some protection against infection, however high this protection may be, it is never perfect, i.e., the efficacy against infection is below 100%. For SARS-CoV-2, the efficacy against infection is generally lower than the efficacy against hospitalization ³⁶.”

The influenza vaccine comment also needs referencing and making more specific. At present it reads that ‘influenza vaccines are not perfect at protecting against infection’. But how? on what evidence?

We added the reference and changed the sentence as follows (Lines 189-190):

“The same holds for influenza vaccines, for which vaccine effectiveness against infection is estimated at 30%-60% ³⁷.”

Where referring to Omicron (e.g I.191) - refrain from saying ‘current VOC’ - BA.1 is already not the current VOC

We removed the reference to Omicron as ‘current VOC’ in the revised MS.

I.191 ‘Vaccination meant that these variants spread faster’ - I think this needs rephrasing. Vaccination and therefore enhanced population immunity has meant that omicron has an

advantage over its predecessor variants with less mutations impacting Nab (so higher comparative growth rate). This does not mean vaccination = faster spread of a virus.

We deleted the sentence as irrelevant at this point.

I. 201 - 'in the longer term, however, mass vaccination further accelerates the rapid evolution of epitope regions'. This is stated as fact, which is inappropriate. It is a fair hypothesis to have and discuss, but should not be stated in this way prior to any discussion of why this statement might be true.

We respectfully disagree. It is not a hypothesis, but a straightforward inference from the well-studied (for other viruses, such as influenza and HIV but also SARS-CoV-2, see Ref. 48) phenomenon of 'antigenic drift' in receptor protein, and the observed fact that a vaccine against SARS-CoV-2 induces protective antibodies. However, we agree that some reference forward is clearly due. Now this passage is placed before the section and reads (Lines 305-309):

"The maximal vaccination coverage supported by public health authorities for controlling the pandemic has important consequences. In the shorter term of a few months, this approach facilitates a decrease in the number of infections, helps to unload hospitals and to reduce COVID-19-related mortality. In the longer term, as we show after two sections, mass vaccination further accelerates the rapid evolution of epitope regions. Let us start, however, with unvaccinated population."

I. 203 - 211. This paragraph does not mention the fact that influenza vaccination happens every year, at least in many countries with temperate climates

We did not mention it in the original MS, because only a very small proportion of the population is vaccinated against seasonal influenza virus, as compared to the mass vaccination of SARS-CoV-2. In the revised MS, we added the following sentence when discussing Figure 4 (Lines 646-649):

"Seasonal influenza with its annual vaccination campaigns in many countries with temperate climates would also correspond to a rather small proportion of vaccinated population, as compared to the global mass vaccination against SARS-CoV-2 realized in the timespan of less than two years."

Fig 2 - please define U_b in the legend as it isn't clear just looking at the figure what 'Ninf U_b ' is

The definition is added to the legend as follows (caption to Figure 3, page 10):

"... mutation rate, U_b , defined as the probability of an escape mutation per transmission in epitopes."

I. 320 - 322. Again, the authors state as fact that vaccination accelerates virus evolution in antibody-binding regions. This can be discussed as a hypothesis but needs to be moderated. The key question is does vaccination increase evolution compared to the alternative scenario - which is waves of infection induced immunity, which will also then wane. The current discussion is very focused around the theoretical concepts of vaccine immunity driving evolution - which is possible. But it can't be discussed as if the alternative is 'no immunity' and therefore no selective pressure. I am not sure that there is enough evidence in what the authors present for vaccination 'favouring immune escape mutations even more' (I.330 - 331)

As we mentioned above acceleration of virus evolution by vaccination is not a hypothesis, but a straightforward logical step from two facts added together. One fact, well-studied and well-cited in our text, is the phenomenon of 'antigenic drift' in variable domains of receptor proteins: Spike (Ref 48), gp120, hemagglutinin. Another well-cited fact is that a vaccine against SARS-CoV-2 induces

neutralizing antibodies that create protection. Because antibodies create protection, regardless of their origin, whether from a natural infection or a vaccine, this protection creates selection pressure for a virus to change and develop resistance. A schematic of the dependence of the substitution rate in epitopes for varying proportions of vaccinated and recovered population that covers a variety of scenarios for Africa, Europe etc is added in new Figure 4, as explained above.

Immunocompromised individuals and the relevance to mutations is mentioned at the end of the manuscript. However, if as is the current hypothesis, most VOC that have led to sweeping replacements globally originated in this way then it is hard to argue that mass vaccination will play any role. It may be that SARSCoV2 evolution settles into a more 'ladder-like' pattern like H3N2 along Omicron lineages (where it would then be more relevant), but we simply do not know yet.

Our perspective is focused on the impact of vaccination on evolution *in rapidly mutating antibody-binding regions* at the population-level (e.g., Abstract, Lines 26-27) and not on the origins of VOC. We explained this in the text (Lines 300-304):

"Below, we focus on the population-level evolution in neutralizing antibody epitopes related to immunity, vaccination, and viral recognition and on escape mutations that occur during transmission chains of acute infections. We postpone until the end the discussion of rare chronic infections, where escape mutations can accumulate within one individual, and of the evolution outside of epitopes."

We do not claim that vaccination will lead to new VOC, we agree on this with the Reviewer. We state very clearly in the text that antigenic drift occurs in epitopes and is unrelated to VOC emergence, which is related to the evolution of the receptor binding domain and other regions (Lines 560-569).

"To avoid confusion, we emphasize that we discuss here the evolution of epitopes only and not, for example, the emergence of VOC related to mutations in other regions. Before Omicron and its descendants, all VOCs had emerged before the mass rollout of vaccination as inferred from phylogenetic analyses^{9,98}. Vaccination was probably not involved as selection pressure in their genesis. The same is likely true for Omicron BA.1."

We are also not interested in predicting the future trajectory of SARS-CoV-2, whether new VOCs will originate or whether, similarly to H3N2, the virus will settle in a ladder-like pattern within Omicron lineages. We agree with the Reviewer on this latter possibility, but this does not interfere with our main message, as we speak everywhere of the average substitution rate in epitopes only, and this ladder exists as oscillations around the average.

Reviewers' comments:

Reviewer #1 (Remarks to the Author):

Overall, nice job on the revision and I am satisfied with how the authors have addressed the comments and suggestions provided during review.

One remaining major point is that the conclusions are overstated somewhat given the theoretical nature of the evidence, thus a final suggestion would be to tone them down a bit. This won't take away from the paper at all just show that the authors aren't over reaching.

A couple small items remaining to attend to:

- L363: "doze" should be dose.

- L794 "About a half of epitopes do not undergo antigenic escape and are left to limit virus replication." This requires a citation.

Reviewer #2 (Remarks to the Author):

Thank you to authors for the revised manuscript. A few of my comments were related to striking the right tone and nuance in a piece like this, which is important in the current climate of public attention on SARSCoV2 vaccination. I think the revised manuscript is much improved in many ways, including this aspect.

I have two minor suggestions:

1. Figure 1 is a nice addition. Can the authors make it clear that these are substitution rates for the whole genome. Could they also consider adding alongside these, the substitution rates for the main antibody target for each virus (e.g. HIV envelope, influenza HA, SARSCoV2 spike)? this would be a valuable visual representation for the reader, especially as in the text the authors mention the 10x higher substitution rate of SARSCoV2 spike compared to flu HA

2. The authors mention T cell responses in the context of preserved effectiveness of prior infection or vaccination against severe infection, as well as mutations within CTL epitopes (l. 660 - 661 and paragraphs that follow). The authors do not discuss the possibility that CTL responses may well enhance viral control and therefore reduce the potential for evolutionary change within an individual. So the analogy of vaccination being akin to a single drug (ART) used in HIV infection is not wholly appropriate. There are several key immunodominant and high conserved CD8 epitopes in non-spike regions and may have a greater impact in areas where a large proportion of the population have been infected.

I appreciate that incorporating the potential impact on T cells of viral control into the models is not possible, but a brief description to acknowledge this potential immune phenomenon could be included.

Reviewer #4 (Remarks to the Author):

Thank you for the opportunity to review this interesting manuscript. It concerns a very important topic – namely the evolution of SARS-CoV-2 in the presence of vaccination – which was discussed widely when vaccines were first introduced. A Perspective piece describing an updated view on this topic, now that SARS-CoV-2 vaccination is widespread, is likely to be a valuable addition to the literature. However, I have concerns about the manuscript in its present form, particularly surrounding how its conclusions (which are based on uncertain assumptions) may be interpreted.

Comments:

1. The numerical arguments presented by the authors in the second part of the manuscript appear to be correct, subject to the underlying assumptions. However, for SARS-CoV-2, there is substantial uncertainty in these underlying assumptions. Are the authors really sure that widespread vaccination is likely to favour the emergence of immune escape variants? Why have most of the variants that we have seen arisen independently of vaccination? I worry that this manuscript will be interpreted as suggesting that mass vaccination is a bad idea, based on a range of assumptions that are far from certain for SARS-CoV-2. I do not contend that the authors' arguments should not be presented, but rather that a more balanced article would also acknowledge the clear possibility that lower case numbers (due to vaccination) *could* reduce, rather than increase, the risk of immune escape, as the level of selection for vaccine escape variants due to vaccination is unknown.
2. In some parts, the manuscript reads in a slightly disjointed way. For example, while a lot of the early text is accurate (about vaccination strategies, and the impacts of host age on transmission), it is not clear how this relates to the authors' main arguments about the impact of vaccination on SARS-CoV-2 evolution. The manuscript would be significantly improved by either deleting much of this text, or making it clearer how the different parts of the manuscript link together (i.e. including the implications for vaccine escape in these earlier sections).
3. There are several places in the manuscript where it seems like the authors are writing as if it is mid-2021. This calls into question whether their conclusions are based on the latest available evidence. The manuscript should be updated. For example: "According to the World Health Organization, the vaccination of the 70% of the world population should be achieved by mid-2022" and "One year into the pandemic, the non-pharmaceutical interventions have been complemented by mass vaccination".
4. There are modelling papers by different authors that come to alternative conclusions about the effects of mass vaccination on the risk of immune escape. See, for example, Gog et al. - Vaccine escape in a heterogeneous population: insights for SARS-CoV-2 from a simple model, and Thompson et al. SARS-CoV-2 incidence and vaccine escape, as well as the Cobey et al. paper that the authors cite. These manuscripts, and other relevant published articles, should be discussed more in this Perspective; evidence suggesting that the immune escape risk may not be increased by mass vaccination should be presented in a balanced way.
5. "The efficacy against infection is below 100%". Could the authors clarify whether they are referring to an assumption of a "leaky" vaccine or "all or nothing" vaccine (i.e. is some proportion of susceptible individuals completely protected, or is the probability of infection per exposure reduced so that many hosts will eventually become infected)? Is this important for the risk of vaccine escape?

6. “The vaccine efficacies or real-world effectiveness against the VOCs generally declined when compared with the Wuhan variant”. Can the authors explain why? Is this because the original vaccines were designed specifically for the initial virus? As written, this makes it sound like vaccine escape has occurred already.

7. Following on from the above, do the authors mention variant-adapted vaccines anywhere? I understand that these have now been deployed to a limited extent so far.

8. Line 142. “the total number of infections did not need to be minimized”. This is simply not true. Having a large number of infections was a worry for policy-makers, aside from the effects on numbers of hospitalisations and deaths. For example, more cases of long COVID, adverse effects on workplaces (due to absences caused by symptomatic but non-hospitalised cases), etc.

9. Lines 148-162. The authors describe models as predictive, and how they allow transmission to be simulated and interventions tested based on biological characteristics of the virus, effectiveness of vaccination etc. While this is true, I think that this over-simplifies things a little. A key benefit of using models is that they allowed uncertainty to be quantified (i.e. they could be run for a range of different assumptions about the factors listed by the authors, rather than those factors being known inputs to model simulations).

10. I am not entirely sure about the authors’ definitions of R_0 and R_e (in Box 2 and in the text). Specifically, what do these quantities assume about public health measures? Doesn’t R_0 assume that there are no control measures in place, whereas R_e accounts for the effects of interventions?

11. Lines 244-248. The authors describe how reinfections can occur due to changes in the virus. But can’t immunity wane due to changes in each host’s immune response too?

12. Line 266. “when product $N_{inf} U_b$ ” should read “when the product $N_{inf} U_b$ ”.

13. “However, as a result of the ongoing evolutionary escape from the immune response after natural infection, as well due to mutations outside of the epitopes including the receptor binding domain, this outcome is altered to the stationary process with seasonal oscillations”. I am not sure that the process is really “seasonal” due to immune escape, is it? Seasonal typically refers to varying based on the season (e.g. winter vs summer) – for example, due to transmission being suppressed as the virus cannot survive in warm weather. Can this please be clarified?

14. Lines 325-329. The authors argue that the main effect of vaccination is to make individuals less infectious. However, wouldn’t vaccination reduce the risk of vaccine escape in multiple ways? For example, it reduces numbers of infections (reducing the opportunity for vaccine escape at the population level), it reduces the viral load (reducing the amount of virus within-host, again reducing the vaccine escape risk). These arguments should be explained in the revised manuscript (again, I think that the authors should present their arguments too and emphasise that immune escape should be considered; but acknowledge uncertainty in whether the immune escape increases or decreases with mass vaccination).

15. Line 333. “orders magnitude” should be “orders of magnitude”? There are quite a few typos like this - these should be checked throughout.

16. Lines 363-365. I understand that V is proportional to $\log(N_{\text{inf}})$, however what is the combined effect of vaccination simultaneously reducing both N_{inf} and U_{b} ?

17. The authors argue against the possibility that the substitution rate is linearly proportional to N_{inf} . However, if vaccine escape mutations with the potential to spread widely are exceptionally rare, and occur with small probability q per infection, then the risk of vaccine escape would surely fall approximately linearly with reducing N_{inf} ?

18. I think that the authors could do a better job of acknowledging the uncertainty in their conclusions. Since the results are based on a range of assumptions about relationships between different variables, do they contend that it is *impossible* that the risk of vaccine escape is lowered by mass vaccination?

19. Relatedly - but less importantly - could a variant that does not respond to vaccination arise by chance rather than due to selection (in which case the risk is lowered by reduced cases due to vaccination)?

20. Caption to Fig 1. Should "monotonously" be "monotonically"?

21. I am a little confused about the claim made in relation to Fig 3a. "In the simplest case, when product $N_{\text{inf}} U_{\text{b}}$ is much less than 1, immune escape mutations emerge and spread through the population one at a time." However, isn't there a possibility that if there are few infections and a low mutation rate, then immune escape mutations simply don't emerge at all (i.e. they take an incredibly long time to appear)? This would be the case in the left part of Fig 3a?

22. The authors argue that Fig 3b applies rather than Fig 3a, based on the substitution rate of SARS-CoV-2. However, given that vaccine escape is a rare event (per infection), isn't U_{b} incredibly small? In that case, I am not convinced that $N_{\text{inf}} U_{\text{b}}$ really is much larger than one.

23. N_{inf} is defined to be the "effective number of infected individuals". The term effective should be defined in this context.

24. Given that we have not yet seen vaccine escape mutations (the authors acknowledge that variants we have seen emerged independently of vaccination), how can we be sure about the selection pressure assumed in the authors' calculations (e.g. equation 2)? While assumptions about the level of selection for immune escape variants may be correct for other viruses, it is not clear that they hold for SARS-CoV-2. The risk of vaccine escape variants emerging involves a balance between vaccination reducing infections (lowering the risk) and possibly increasing selection for immune escape variants (potentially increasing the risk; but the extent to which this occurs is unknown).

In summary: As described above, I think the authors could turn this into a nice Perspective piece that comes to a similar conclusion - that immune escape should be considered when designing vaccines/vaccination strategies - but I think that substantial work is required to make this manuscript more balanced and to avoid misinterpretation.

Reviewer #1

Overall, nice job on the revision and I am satisfied with how the authors have addressed the comments and suggestions provided during review.

Thank you for your assessment.

One remaining major point is that the conclusions are overstated somewhat given the theoretical nature of the evidence, thus a final suggestion would be to tone them down a bit. This won't take away from the paper at all just show that the authors aren't over reaching.

Thank you. We have toned down our conclusions by adding a paragraph, as follows (Lines 226-236):

“In what follows, we assume – and this is the only essential assumption on which our discussion relies – that the mutational cost of mutations in antibody-neutralizing regions for SARS-CoV-2 is as small as for influenza virus and HIV. We make this assumption because the structure of antibody-binding sites on spike is similar to gp120 of HIV and hemagglutinin of influenza. It comprises several protrusions covered in sugars, located far from the receptor binding site and serving as targets for antibodies. The selection pressure for the virus to escape is of the same order of magnitude as well. It is given by the inverse number of “cheap” recognized sites in the epitopes, which for these viruses are in the same range of lengths. Although we cannot guarantee that this assumption is true, it is likely, and we hope that further experiments will test it. Most research has so far focused on mutations causing the emergence of VOC, and we hope that our Perspective will bring more focus to finding new epitope variants for SARS-CoV-2 as it happened for influenza and HIV.”

A couple small items remaining to attend to:

- L363: "doze" should be dose.

Fixed.

- L794 "About a half of epitopes do not undergo antigenic escape and are left to limit virus replication." This requires a citation.

Added {Ganusov, 2011;Goonetilleke, 2009}

Reviewer #2:

Thank you to authors for the revised manuscript. A few of my comments were related to striking the right tone and nuance in a piece like this, which is important in the current climate of public attention on SARSCoV2 vaccination. I think the revised manuscript is much improved in many ways, including this aspect.

Thank you for your assessment.

I have two minor suggestions:

1. Figure 1 is a nice addition. Can the authors make it clear that these are substitution rates for the whole genome.

We have made this clear in the figure caption.

Could they also consider adding alongside these, the substitution rates for the main antibody target for each virus (e.g. HIV envelope, influenza HA, SARSCoV2 spike)? this would be a valuable visual representation for the

reader, especially as in the text the authors mention the 10x higher substitution rate of SARSCoV2 spike compared to flu HA

We have made this addition in the figure and made it clear in the figure caption (see above).

2. The authors mention T cell responses in the context of preserved effectiveness of prior infection or vaccination against severe infection, as well as mutations within CTL epitopes (l. 660 - 661 and paragraphs that follow). The authors do not discuss the possibility that CTL responses may well enhance viral control and therefore reduce the potential for evolutionary change within an individual. So the analogy of vaccination being akin to a single drug (ART) used in HIV infection is not wholly appropriate.

The number of drugs or responses is not important for our analogy at all, we just cited another example of suboptimal therapy with pre-existing mutants. To avoid further confusion, we simplified the passage to (Lines 391-394):

“This effect is analogous to the case of suboptimal therapy in an HIV-infected individual that selects for drug-resistant mutants. These mutants exist in very small quantities before therapy and become dominant in a patient within weeks of failing therapy. Highly-active drug cocktails have solved this problem.”

There are several key immunodominant and high conserved CD8 epitopes in non-spike regions and may have a greater impact in areas where a large proportion of the population have been infected.

We agree on this point.

I appreciate that incorporating the potential impact on T cells of viral control into the models is not possible, but a brief description to acknowledge this potential immune phenomenon could be included.

In the original manuscript, we discussed the CTL response at the end (Lines 532-549). In the revised manuscript, we have added (Lines 533-536):

“CTL immune response in a host lowers viral load and hence the transmission rate at the population level. This effect is incorporated in the reproduction number, R_0 , which affects the selection pressure of antigenic escape, as given by Equation 1.”

Reviewer #4:

Thank you for the opportunity to review this interesting manuscript. It concerns a very important topic – namely the evolution of SARS-CoV-2 in the presence of vaccination – which was discussed widely when vaccines were first introduced. A Perspective piece describing an updated view on this topic, now that SARS-CoV-2 vaccination is widespread, is likely to be a valuable addition to the literature.

Thank you for finding our manuscript interesting and recognizing the importance of the topic.

However, I have concerns about the manuscript in its present form, particularly surrounding how its conclusions (which are based on uncertain assumptions) may be interpreted.

We understand Reviewer's concerns and have clarified them in our responses below.

Comments:

1. a) The numerical arguments presented by the authors in the second part of the manuscript appear to be correct, subject to the underlying assumptions. [...] Are the authors really sure that widespread vaccination is likely to favour the emergence of immune escape variants?

18. I think that the authors could do a better job of acknowledging the uncertainty in their conclusions. Since the results are based on a range of assumptions about relationships between different variables, do they contend that it is *impossible* that the risk of vaccine escape is lowered by mass vaccination?

We cited other studies [Gog et al 2021; Thompson et al 2021; Cobey et al 2021; Koelle et al 2006] (Lines 307, 430, 501) and explained why they make conclusions different from ours, regardless of the validity of our own assumptions. The main point is that these studies do not use the modern theory of antigenic evolution, but use some simplified infectious disease modeling approaches or arguments based on them (Lines 430-442).

In contrast, in this Perspective, we use the results of the modern multi-locus theory applied to the problem of antigenic response by three leading groups in modeling viral evolution under rather different assumptions [Rouzine et al 2018; Yan et al 2019; Marchi et al 2021]. Equation 1 was shown to be robust to their diverse assumptions and to fit earlier experimental estimates and

simulations for the influenza virus, which has a very similar biochemical structure of antibody-binding regions that evolved to avoid virus neutralization. The impact of vaccination on the antigenic evolution of SARS-CoV-2 is also recognized in observational studies, see [Yewdell, 2021] we cited in Lines 262-264.

To summarize our point of view, we have added the following sentence before the Future Research section (Lines 500-503):

“To summarize, we arrived at very different conclusions regarding the effect of vaccination on the speed of antigenic evolution compared to the previous work ^{75,77,78} by exploiting the similarities in the molecular structure of antibody-binding regions of different viruses and using the modern theory of multi-locus evolution driven by immune response.”

1. b) However, for SARS-CoV-2, there is substantial uncertainty in these underlying assumptions. [...]

[...] based on a range of assumptions that are far from certain for SARS-CoV-2.

In fact, the only essential assumption made in our Perspective is that antibody-binding regions of Spike of SARS-CoV-2 are as easy to mutate as in gp120 of HIV and hemagglutinin of influenza virus. The assumption is made due to the similarity in the protein structure: several protrusions far from functional binding domain covered with sugars that serve as decoys/targets for antibodies. (This is in striking contrast to CTL epitopes that are randomly located and cannot evolve in this way.)

To stress that this is the only essential assumption made in our Perspective, we have added a paragraph (Lines 226-236):

“In what follows, we assume – and this is the only essential assumption on which our discussion relies – that the mutational cost of mutations in antibody-neutralizing regions for SARS-CoV-2 is as small as for influenza virus and HIV. We make this assumption because the structure of antibody-binding sites on spike is similar to gp120 of HIV and hemagglutinin of influenza. It comprises several protrusions covered in sugars, located far from the receptor binding site and serving as targets for antibodies. The selection pressure for the virus to escape is of the same order of magnitude as well. It is given by the inverse number of “cheap” recognized sites in the epitopes, which for these viruses are in the same range of lengths. Although we cannot guarantee that this assumption is true, it is likely, and we hope that further experiments will test it. Most research has so far focused on

mutations causing the emergence of VOC, and we hope that our Perspective will bring more focus to finding new epitope variants for SARS-CoV-2 as it happened for influenza and HIV.”

1. c) Why have most of the variants that we have seen arisen independently of vaccination?

Because the genome is much longer than the total length of the antibody epitopes. Besides, very few people have been looking for antigenic variants yet. We already cited Yewdell et al 2021, as one of the exceptions.

The SARS-CoV-2 pandemic is young, data are still accumulating, and researchers have more pressing concerns. Everyone is focused on virulence and mutations causing emergence of VOC, which arise due to the evolution of genomic regions unrelated to vaccines, such as the receptor-binding domain. There are hundreds of mutating sites, so it is like randomly finding a needle in a hay stack. One cannot find a phenotype unless one designs an expensive experiment. We hope that our Perspective will change this situation and serve as a pointer to epitope variants. To explain this view, we added the following sentence in the revised manuscript (Lines 234-236):

“Most research has so far focused on mutations causing the emergence of VOC, and we hope that our Perspective will bring more focus to finding new epitope variants for SARS-CoV-2 as it happened for influenza and HIV.”

1. d) I worry that this manuscript will be interpreted as suggesting that mass vaccination is a bad idea [...]

Nowhere in the manuscript we claimed that vaccination is a bad idea. On the contrary, a significant part of this manuscript is focused on how vaccination was instrumental in achieving the control of the virus and preventing deaths. Our point in considering the impact of vaccination on antigenic evolution of SARS-CoV-2 is that we should work together to identify possible solutions. For influenza virus, large collaborations exist to monitor antigenic evolution. We hope for SARS-CoV-2 further research will include a variety of promising approaches outlined in the discussion.

1. e) I do not contend that the authors' arguments should not be presented, but rather that a more balanced article would also acknowledge the clear possibility that lower case numbers (due to vaccination) *could* reduce, rather than increase, the risk of immune escape, as the level of selection for vaccine escape variants due to vaccination is unknown.

Actually, there is an entire scientific field that focuses on successful prediction of the parameters of virus evolution based on data, and we cite some of these papers (Rouzine & Coffin 1999 and 2005, Rouzine et al 2003, Desai and Fisher 2007, Neher et al 2010, Batorsky et al 2011, Schiffels et al 2011, Good et al 2012, Neher et al 2013, Desai et al 2013). The topic has been studied rather intensely for similar systems for 20 years. To address the Reviewer's concern, without diluting the message of our article, we have added a paragraph with a disclaimer at the beginning of the evolution section (Lines 226-236, see our response to point 1. b).

2. In some parts, the manuscript reads in a slightly disjointed way. For example, while a lot of the early text is accurate (about vaccination strategies, and the impacts of host age on transmission), it is not clear how this relates to the authors' main arguments about the impact of vaccination on SARS-CoV-2 evolution. The manuscript would be significantly improved by either deleting much of this text, or making it clearer how the different parts of the manuscript link together (i.e. including the implications for vaccine escape in these earlier sections).

The 'early' text is written for researchers with the background in viral evolution and general readers without specific knowledge of infectious disease modeling and epidemiology. We see this part as needed and far from being trivial for researchers other than those familiar with infectious disease modeling and adjacent fields. The intent of this important part, which takes only 3 pages, is to present an introduction and to make clear why we need vaccination in the first place and how it has been instrumental in controlling the pandemic, which is precisely what the Reviewer worries about.

3. There are several places in the manuscript where it seems like the authors are writing as if it is mid-2021. This calls into question whether their conclusions are based on the latest available evidence. The manuscript should be updated. For example: "According to the World Health Organization, the vaccination of the 70% of the world population should be achieved by mid-2022" and "One year into the pandemic, the non-pharmaceutical interventions have been complemented by mass vaccination".

We have updated the text throughout to reflect the current situation.

4. There are modelling papers by different authors that come to alternative conclusions about the effects of mass vaccination on the risk of immune escape. See, for example, Gog et al. - Vaccine escape in a heterogeneous population: insights for SARS-CoV-2 from a simple model, and Thompson et

al. SARS-CoV-2 incidence and vaccine escape, as well as the Cobey et al. paper that the authors cite. These manuscripts, and other relevant published articles, should be discussed more in this Perspective; evidence suggesting that the immune escape risk may not be increased by mass vaccination should be presented in a balanced way.

We appreciate that the Reviewer brings these articles to our attention. Gog et al 2021 and Thompson et al 2021 are now cited in the revised manuscript, in addition to Cobey et al 2021 that was already included in the original manuscript. We realize that the conclusions of these important studies are at the core of Reviewer's concerns because they differ from our conclusions.

We would like to point out that all these studies use rather standard compartmental infectious disease models and arguments following from those to show that vaccination can decrease vaccine escape through reducing the number of infectious individuals. Our key observation included in the revised text is that the aforementioned studies lack evolutionary dynamics as such, it is just not built into these simplified infectious disease models. For example, in Gog et al 2021, the vaccine escape pressure is merely assumed to be proportional to the number of vaccinated individuals. At best, models of this type correspond to the case when immune escape mutations emerge and spread through the population one at a time (so-called single-site approximation, see Figure 3A). The fact that this approximation is too crude is an important point in our Perspective. Note that, in the seminal Science 2004 paper by Bryan Grenfell et al "Unifying epidemiological and evolutionary dynamics of pathogens", this limitation was acknowledged, however it is completely neglected in the majority of compartmental infectious disease models that aim to address the question of vaccine escape and impact of vaccination on viral antigenic evolution. Infectious disease models ignore proper treatment of evolutionary dynamics and disregard mutations at multiple sites, because such models are technically more difficult to analyze. At the same time, it is well-known (Strelkova and Lassig, 2012; Batorsky et al 2012) that, for influenza, HIV and other viruses, mutations at different positions of the viral genome are concurrent in time, and that the approximation of independent mutations does not apply anymore due to strong interference between different mutations.

In other words, in our Perspective, we indirectly say that simplified epidemiological models like those in Gog et al 2021 disregard important evolutionary effects that drastically alter the conclusions about the impact of vaccination on antigenic evolution – the effects that have been demonstrated in theoretical and bioinformatical studies and accepted by evolutionary theorists since decades. These are true for influenza viruses, but also other viruses which have strong antigenic evolution like SARS-CoV-2. Therefore,

we see our role in this Perspective to bridge the two fields of infectious disease dynamics and evolutionary theory. As pointed out by Grenfell et al 2004, simple one-locus models ignore many factors, such as varying population size, clonal interference, and epistatic interactions (see their Ref 29 and Fig 2A for an individual-host, similar to our Figure 3A for a population level model).

We have now added an explanation of how it is related to the prior infectious disease modeling work, as follows (Lines 430-442):

“Several studies ^{75,77,78} use standard compartmental infectious disease models and arguments following from those to predict that vaccination can decrease vaccine escape by reducing the number of infectious individuals. We would like to point out that the aforementioned studies lack evolutionary dynamics because it is not built into these simplified models. For example, Gog and colleagues ⁷⁷ assume the vaccine escape pressure to be proportional to the number of vaccinated individuals. At best, models of this type correspond to the case when immune escape mutations emerge and spread through the population one at a time (single-site approximation; Figure 3a). As we explained above, this is the case for unrealistically-small population sizes. These models do not include the proper treatment of evolutionary dynamics and disregard linkage effects existing between mutations at multiple sites, because such models are technically more difficult to handle. In this Perspective, we use the modern theory of multi-locus virus evolution that takes into account clonal interference, genetic background effect and other linkage effects, random genetic drift, and the natural selection arising due to immune memory.”

5. “The efficacy against infection is below 100%”. Could the authors clarify whether they are referring to an assumption of a “leaky” vaccine or “all or nothing” vaccine (i.e. is some proportion of susceptible individuals completely protected, or is the probability of infection per exposure reduced so that many hosts will eventually become infected)? Is this important for the risk of vaccine escape?

Clarified at the relevant place (Lines 395-398):

“Note that since we consider the dynamics at the population level, the effect discussed here will be the same both for a “leaky” vaccine, where all susceptible individuals have reduced susceptibility to infection after vaccination, and for an “all or nothing” vaccine, where a proportion of susceptible individuals are completely protected by vaccination.”

6. “The vaccine efficacies or real-world effectiveness against the VOCs generally declined when compared with the Wuhan variant”. Can the authors explain why? Is this because the original vaccines were designed specifically for the initial virus? As written, this makes it sound like vaccine escape has occurred already.

We added (Lines 135-137):

“The decline in vaccine efficacy to decrease the number of infections was partly because the original vaccines were designed specifically for the initial virus, and, partly, due to the immune escape detailed below. At the moment, we cannot quantify which effect is larger.”

7. Following on from the above, do the authors mention variant-adapted vaccines anywhere? I understand that these have now been deployed to a limited extent so far.

In the future strategy section, we added now (Line 137-140):

“Bivalent mRNA vaccines adapted to target the original variant and a more recent Omicron subvariant have now been approved and deployed to a limited extent, similar to influenza vaccines that are matched to the most prevalent circulating variant.”

8. Line 142. “the total number of infections did not need to be minimized”. This is simply not true. Having a large number of infections was a worry for policy-makers, aside from the effects on numbers of hospitalisations and deaths. For example, more cases of long COVID, adverse effects on workplaces (due to absences caused by symptomatic but non-hospitalised cases), etc.

The situation was dependent on the country. In some countries like China, the policy was very different. In many other countries, including the Netherlands where one of us works, there was no goal whatsoever to minimize the number of infections. Not to say that, of course, there are consequences of having a non-hospitalized SARS-CoV-2 infection like long COVID etc, these were not taken into consideration by policymakers. We replaced the sentences

“In contrast, the total number of infections did not need to be always minimized. Since a large proportion of infections are asymptomatic or mild, they do not require any medical attention and do not disrupt the social and economic activity.”

with (Lines 145-149)

“The situation regarding minimizing the total number of infections differed by country. In some countries like China, zero COVID-19 policy was applied. In other countries, including the Netherlands, consequences of non-hospitalized SARS-CoV-2 infection such as long COVID and work absenteeism were recognized, but still, there was no goal to minimize the total number of infections.”

9. Lines 148-162. The authors describe models as predictive, and how they allow transmission to be simulated and interventions tested based on biological characteristics of the virus, effectiveness of vaccination etc. While this is true, I think that this over-simplifies things a little. A key benefit of using models is that they allowed uncertainty to be quantified (i.e. they could be run for a range of different assumptions about the factors listed by the authors, rather than those factors being known inputs to model simulations).

We added (Lines 292-294):

“Mathematical models connect the initial assumptions to predictions in the most accurate and reproducible way. A key benefit of using models is that they allow uncertainty to be quantified and to conduct scenario analyses based on a range of initial assumptions.”

10. I am not entirely sure about the authors' definitions of R_0 and R_e (in Box 2 and in the text). Specifically, what do these quantities assume about public health measures? Doesn't R_0 assume that there are no control measures in place, whereas R_e accounts for the effects of interventions?

No. By the definition in the cited models, lockdowns affect both R_0 and R_e equally. We clarified now (Lines 273-276):

“Both R_0 and R_e , are defined for given public measures in place, such as lockdowns and various restrictions. They differ, by definition, only due to the immune memory accumulated in a population. Therefore, public health measures reduce R_0 and R_e by the same factor.”

11. Lines 244-248. The authors describe how reinfections can occur due to changes in the virus. But can't immunity wane due to changes in each host's immune response too?

Added in line 214-215:

“, in addition to waning of antibodies,”

12. Line 266. “when product $N_{inf} U_b$ ” should read “when the product $N_{inf} U_b$ ”.

Corrected (Line 305 and 309).

13. “However, as a result of the ongoing evolutionary escape from the immune response after natural infection, as well due to mutations outside of the epitopes including the receptor binding domain, this outcome is altered to the stationary process with seasonal oscillations”. I am not sure that the process is really “seasonal” due to immune escape, is it? Seasonal typically refers to varying based on the season (e.g. winter vs summer) – for example, due to transmission being suppressed as the virus cannot survive in warm weather. Can this please be clarified?

Clarified (Line 284-285):

“...with seasonal oscillations in the virus prevalence due to seasonality in transmission”

14. Lines 325-329. The authors argue that the main effect of vaccination is to make individuals less infectious. However, wouldn't vaccination reduce the risk of vaccine escape in multiple ways? For example, it reduces numbers of infections (reducing the opportunity for vaccine escape at the population level), it reduces the viral load (reducing the amount of virus within-host, again reducing the vaccine escape risk). These arguments should be explained in the revised manuscript (again, I think that the authors should present their arguments too and emphasize that immune escape should be considered; but acknowledge uncertainty in whether the immune escape increases or decreases with mass vaccination).

We addressed this issue in the revised manuscript in several places. Despite of the decrease of mutation rate U_b due to the decrease of viral load by vaccine, such a reduction also **increases** the selection pressure to virus to escape, not decreases it. To avoid further confusion, we now explain in more detail why the second effect wins, as follows (Lines 425-430):

“Thus, vaccination creates two opposing effects on Red Queen adaptation in epitopes. One effect comes from the decrease in transmission rate due to partial immune protection and lower virus amount transmitted to another individual. This effect creates a positive selection pressure for resistant mutations. The opposite effect of vaccination comes from the decrease in the mutation rate within a host, due to lower viral load. The first effect wins,

because the adaptation rate is linearly proportional to selection pressure, s , and weakly depends on the mutation rate, U_b (Figure 3b)."

15. Line 333. "orders magnitude" should be "orders of magnitude"? There are quite a few typos like this - these should be checked throughout.

The typo is fixed. Thank you.

16. Lines 363-365. I understand that V is proportional to $\log(N_{\text{inf}})$, however what is the combined effect of vaccination simultaneously reducing both N_{inf} and U_b ?

Clarified (Lines 415-416):

"As already mentioned, all multi-locus models demonstrate that V depends logarithmically on both N_{inf} and mutation rate, U_b (Rouzine 2020, book "Mathematical modeling of evolution", De Gruyter, Berlin/Boston)."

17. The authors argue against the possibility that the substitution rate is linearly proportional to N_{inf} . However, if vaccine escape mutations with the potential to spread widely are exceptionally rare, and occur with small probability q per infection, then the risk of vaccine escape would surely fall approximately linearly with reducing N_{inf} ?

21. I am a little confused about the claim made in relation to Fig 3a. "In the simplest case, when product $N_{\text{inf}} U_b$ is much less than 1, immune escape mutations emerge and spread through the population one at a time." However, isn't there a possibility that if there are few infections and a low mutation rate, then immune escape mutations simply don't emerge at all (i.e. they take an incredibly long time to appear)? This would be the case in the left part of Fig 3a?

22. The authors argue that Fig 3b applies rather than Fig 3a, based on the substitution rate of SARS-CoV-2. However, given that vaccine escape is a rare event (per infection), isn't U_b incredibly small? In that case, I am not convinced that $N_{\text{inf}} U_b$ really is much larger than one.

If mutants were "incredibly rare", we would not observe hundreds of diverse sites in SARS-CoV-2 and the emergence of VOC twice a year. In fact, mutants are not "exceptionally rare", because RNA viruses lack proofreading enzymes.

Added (Lines 321-329):

“Fast evolution is common for RNA viruses. Because they lack proofreading enzymes, their mutation rates are relatively large¹⁰¹. They all fall in the range of 10^{-6} to 10^{-4} per nucleotide per replication. To estimate U_b , we have to multiply this mutation rate by the size of infected population and the length of antibody binding region. For influenza A, the mutation rate per transmission event per antibody binding region is estimated at 3×10^{-4} ^{61,70}. For SARS-CoV-2, the population-level mutation rate is expected to fall within the same order of magnitude. Therefore, the multi-locus regime ($N_{inf} U_b > 1$) applies if more than $N_{inf} = 10,000$ infected individuals are present in a population, which is the case during a pandemic wave in a large city. The fact that $N_{inf} U_b > 1$ for influenza A H2N3, which falls into the clonal interference regime, was demonstrated using sequence data⁵⁴.”

19. Relatedly - but less importantly - could a variant that does not respond to vaccination arise by chance rather than due to selection (in which case the risk is lowered by reduced cases due to vaccination)?

A new genetic variant always arises by chance mutation, but is subsequently amplified (or suppressed) by natural selection, with random genetic drift on the top. These factors of evolution act all together, in synergy. To avoid further confusion, we added (Lines 333-334):

“Genetic variants arise by random mutation but are subsequently amplified or suppressed by natural selection, with random genetic drift and linkage as additional stochastic factors.”

20. Caption to Fig 1. Should “monotonously” be “monotonically”?

Fixed.

23. N_{inf} is defined to be the “effective number of infected individuals”. The term effective should be defined in this context.

We removed “effective”, for simplicity.

24. Given that we have not yet seen vaccine escape mutations

See our response to Comment 1 above.

(the authors acknowledge that variants we have seen emerged independently of vaccination), how can we be sure about the selection pressure assumed in the authors’ calculations (e.g. equation 2)?

It is not an “assumption” but a well-known result obtained independently by the groups working in virus evolution, as discussed in the detail in the text.

While assumptions about the level of selection for immune escape variants may be correct for other viruses, it is not clear that they hold for SARS-CoV-2.

Not clear, unless we know something about their virology that makes them similar (see our new paragraph in Lines 321-329 mentioned above).

The risk of vaccine escape variants emerging involves a balance between vaccination reducing infections (lowering the risk) and possibly increasing selection for immune escape variants (potentially increasing the risk; but the extent to which this occurs is unknown).

We now discuss this balance in more detail (Lines 425-430):

“Thus, vaccination creates two opposing effects on Red Queen adaptation in epitopes. One effect comes from the decrease in transmission rate due to partial immune protection and lower virus amount transmitted to another individual. This effect creates a positive selection pressure for resistant mutations. The opposite effect of vaccination comes from the decrease in the mutation rate within a host, due to lower viral load. The first effect wins, because the adaptation rate is linearly proportional to selection pressure, s , and weakly depends on the mutation rate, U_b (Figure 3b).”

In summary: As described above, I think the authors could turn this into a nice Perspective piece that comes to a similar conclusion - that immune escape should be considered when designing vaccines/vaccination strategies - but I think that substantial work is required to make this manuscript more balanced and to avoid misinterpretation.

We hope that the above corrections and clarifications helped to address this concern.

REVIEWERS' COMMENTS:

Reviewer #2 (Remarks to the Author):

Thank you, my comments have been addressed

Reviewer #4 (Remarks to the Author):

Thanks to the authors for responding to the comments in my original review. In general, they have done a good job at addressing my queries.

My remaining uncertainty, however, lies in my worry that this Perspective will be misinterpreted as arguing against mass vaccination. I am aware that the authors do not explicitly say this, but their Perspective does suggest (without acknowledging the substantial uncertainty around this clearly enough) that global vaccination will accelerate SARS-CoV-2 evolution. While this *may* be true, I do not think that this can be stated unambiguously without empirical evidence from the SARS-CoV-2 variants that have emerged in the real world, however sophisticated the authors' theoretical argument is. Most new SARS-CoV-2 variants have emerged prior to the widespread use of vaccines. As such, the fact that equation 1 has been shown to hold for flu is not really sufficient to say that the authors' theory is correct. I remain to be convinced by the authors (due to the limited empirical evidence) that SARS-CoV-2 vaccination definitely does exert a substantial selection pressure for resistant variants. I therefore think that the strength of the conclusions here need to be toned down.

In my original review, I did not intend the authors to revise their manuscript to suggest that their approach is superior to work by Gog and others. Instead, they should do a much better job of acknowledging that their own work relies on assumptions that may not be valid (including uncertainty in the conceptual model, as well as its inputs) - and that this work is intended to demonstrate that alternative conclusions about the effects of vaccination can be reached using a different framework compared to earlier work (rather than arguing so strongly that the earlier work was too basic, and that the conclusions here are definitely correct).

Since the authors have chosen not to present the possibility that a range of conclusions can be reached - and have instead chosen to argue fully for the results emerging from their model (however sophisticated it may be), I am afraid that I cannot support publication of this Perspective. As I said previously, a substantial change in the tone of the Perspective could certainly allow this work to be a nice addition to the literature.

Even the addition of a sentence expressing uncertainty at the end of the abstract would help (with similar phrasing changes later in the manuscript). For example, they could replace the last sentence of the abstract with: "Of course, this is purely theoretical and so this conclusion is far from certain. However, if it holds, we discuss the potential implications of this phenomenon for viral dynamics and future vaccination strategies."

Editors

Thank you once again for the opportunity to revise our manuscript. The list of requested changes is given below. The Lines indicate changes in the marked manuscript.

In line with Reviewer 4's comments, please provide some expression of uncertainty around your conclusions in the Abstract.

We edited the abstract as follows (Lines 23-31):

Mass vaccination was the main pillar of the public health response to the COVID-19 pandemic. It was very effective in reducing hospitalizations and deaths. At the same time, SARS-CoV-2 may escape from both natural and vaccine-induced immunity. We provide a perspective in the context of the viral evolutionary theory on how vaccination might accelerate SARS-CoV-2 evolution in antibody-binding regions compared to natural infection, at the population level. Using the evidence of strong genetic variation in antibody-binding regions and taking advantage of the similarity between the envelope proteins of SARS-CoV-2 and influenza, we assume that immune selection pressure acting on these regions of the two viruses is similar and discuss existing models of influenza evolution. We further outline the implications of this phenomenon for future vaccination strategies.

Replace with 'vaccination can also have consequences...'

We edited this sentence as follows (Lines 64-65):

However, the pillar of the public health response to COVID-19, vaccination can also have consequences for SARS-CoV-2 evolution in antibody-binding regions located in spike protein.

This sentence could be toned down to clarify that vaccination *might* or *potentially* increases selective pressure.

As requested, the sentence was edited (Lines 68-71):

Mass vaccination, as we show below, might increase this pressure and accelerate SARS-CoV-2 evolution in spike epitopes compared to natural infection.

Author, this sentence should also be toned down to explain that this is not an absolute certainty.

We added “potentially” to this sentence (Lines 250-252):

In the longer term, as we show in this Perspective, mass vaccination potentially further accelerates the rapid evolution of epitope regions.

Author, please see comments from Reviewer 4 on this paragraph. We understand from your rebuttal why you wish to explain how these prior

studies differ from your work, but please ensure it is clear that your own work is built on assumptions that may be uncertain.

We continued this paragraph as follows (Lines 453-467):

From the evolutionary viewpoint, multi-locus models are closer to reality than independent-locus models. At the same time, based on strong genetic variation in the antibody epitopes in spike of SARS-CoV-2, we assumed that mutations in these regions have a low cost, by analogy with mutations in hemagglutinin protein of influenza. The validity of these assumptions remains to be tested directly in the future.

Author, some sentences such as this one, and the first sentence in the next section ('The vaccine design that does not...') suggest that it is definitively and empirically proven that current designs accelerate virus evolution too much. The overall message on the need to consider viral evolution in vaccine design can remain, but please revise these sentences to make it clear that this is not necessarily a certainty.

We edited the sentences as follows (Lines 527-530 and 532):

If SARS-CoV-2 continues to cause substantial burden of severe disease in vulnerable individuals, we should either design a type of vaccine that does not carry any potential danger of accelerating virus evolution in epitopes but is still effective against severe disease, or find other methods of reducing virus circulation.

The vaccine design that prevents all virus evolution in antibody-binding regions is not obvious.

Potentially.

This sentence was edited (Lines 549-550):

This leads to the need for repeated rounds of vaccination potentially further accelerating virus evolution.

Please clarify what you mean by monotonous here.

The clarification was added (Lines 601-602):

Various models suggest that the evolution towards the increase of virulence does not need to be monotonous in the sense of continuing virulence decrease¹²⁶⁻¹²⁸.

Author, we very much understand from the text and your rebuttal that you are not intending to argue against mass vaccination, rather to improve vaccine design in light of viral evolution.

However, we understand Reviewer 4's concerns that the text must be carefully caveated to explain uncertainty around your conclusions. Please add some text to your conclusions here in line with previous comments.

Please replace with 'This possibility...' or a similar wording.

We edited the conclusions as follows (Lines 673-689):

Mass vaccination has been very effective in reducing deaths, severe disease and overall disease burden due to COVID-19 in many countries. At the same time, SARS-CoV-2 may escape from both natural and vaccine-induced immunity. In this Perspective, we discussed the possibility that global vaccination may accelerate SARS-CoV-2 evolution in rapidly mutating antibody-binding regions compared to natural infection. Our conclusions rely on the assumption that immune selection pressure acting on the antibody-binding regions of SARS-CoV-2 is similar to that of influenza, and on existing multi-locus models of influenza evolution. For this end, we took advantage of the similarity between the envelope proteins of the two viruses and the evidence of strong genetic variation in the antibody epitopes. The validity of our assumption remains to be tested directly for SARS-CoV-2 in the future, as it was for done for influenza virus. The potential impact of vaccination on SARS-CoV-2 evolution should be acknowledged for future vaccination strategies that target most at-risk populations, especially if vaccination campaigns will cover a substation part of the population. Mutations in immunologically-relevant genomic regions, viral recombination, virulence and fitness evolution must be considered when designing a future vaccination strategy. Finally, we would like to stress that despite potential implications of vaccination for evolution in the antibody epitopes, in face of an unprecedented global health crisis like the one we just experienced, mass vaccination is probably the only tool to prevent widespread loss of human lives and huge economic costs.

Reviewer #4 (Remarks to the Author):

Thanks to the authors for responding to the comments in my original review. In general, they have done a good job at addressing my queries.

We are glad with this assessment of our response to the Reviewer's comments. The remaining comments are addressed below. Please note that the lines indicate changes in the marked manuscript.

My remaining uncertainty, however, lies in my worry that this Perspective will be misinterpreted as arguing against mass vaccination.

To make our vision clear, we have added a sentence in conclusions (Lines 686-689):

Finally, we would like to stress that despite potential implications of vaccination for evolution in the antibody epitopes, in face of an unprecedented global health crisis like the one we just experienced, mass vaccination is probably the only tool to prevent widespread loss of human lives and huge economic costs.

I am aware that the authors do not explicitly say this, but their Perspective does suggest (without acknowledging the substantial uncertainty around this clearly enough) that global vaccination will accelerate SARS-CoV-2 evolution. While this *may* be true, I do not think that this can be stated unambiguously without empirical evidence from the SARS-CoV-2 variants that have emerged in the real world, however sophisticated the authors' theoretical argument is. Most new SARS-CoV-2 variants have emerged prior to the widespread use of vaccines.

We believe there is still a misunderstanding here. Nowhere we claimed that vaccination causes the emergence of VoC, but we rather explained the contrary several times, and note that the emergence of VoC is not at all what we discuss in our perspective. We discuss that vaccination may lead to new antigenic variants (which is not the same as another VoC, and evolution of VoC is not the same as antigenic evolution of SARS-CoV-2 that we focus on). The antigenic evolution for SARS-CoV-2 is an inherent part of the virus biology, see, e.g., other comments on this topic and references below. The second reference is one of the first empirical confirmations of the existence of new antigenic variants of SARS-CoV-2 (precisely as it was shown for flu before).

Markov, P.V., Katzourakis, A. & Stilianakis, N.I. Antigenic evolution will lead to new SARS-CoV-2 variants with unpredictable severity. *Nat Rev Microbiol* **20**, 251–252 (2022).
<https://doi.org/10.1038/s41579-022-00722-z>

Mapping SARS-CoV-2 antigenic relationships and serological responses

Samuel H. Wilks, Barbara Mühlemann, Xiaoying Shen, Sina Türeli, Eric B. LeGresley, Antonia Netzl, Miguela A. Caniza, Jesus N. Chacaltana-Huarcaya, Victor M. Corman, Xiaoju Daniell, Michael B. Datto, Fatimah S. Dawood, Thomas N. Denny, Christian Drosten, Ron A. M. Fouchier, Patricia J. Garcia, Peter J. Halfmann, Agatha Jassem, Lara M. Jeworowski, Terry C. Jones, Yoshihiro Kawaoka, Florian Krammer, Charlene McDanal, Rolando Pajon, Viviana Simon, Melissa S. Stockwell, Haili Tang, Harm van Bakel, Vic Vegaulla, Richard Webby, David C. Montefiori, Derek J. Smith

bioRxiv 2022.01.28.477987; doi: <https://doi.org/10.1101/2022.01.28.477987>

As such, the fact that equation 1 has been shown to hold for flu is not really sufficient to say that the authors' theory is correct. I remain to be convinced by the authors (due to the limited empirical evidence) that SARS-CoV-2 vaccination definitely does exert a substantial selection pressure for resistant variants. I therefore think that the strength of the conclusions here need to be toned down.

We edited the conclusions as follows (Lines 673-689):

Mass vaccination has been very effective in reducing deaths, severe disease and overall disease burden due to COVID-19 in many countries. At the same time, SARS-CoV-2 may escape from both natural and vaccine-induced immunity. In this Perspective, we discussed the possibility that global vaccination may accelerate SARS-CoV-2 evolution in rapidly mutating antibody-binding regions compared to natural infection. Our conclusions rely on the assumption that immune selection pressure acting on the antibody-binding regions of SARS-CoV-2 is similar to that of influenza, and on existing multi-locus models of influenza evolution. For this end, we took advantage of the similarity between the envelope proteins of the two viruses and the evidence of strong genetic variation in the antibody epitopes. The validity of our assumption remains to be tested directly for SARS-CoV-2 in the future, as it was for done for influenza virus. The potential impact of vaccination on SARS-CoV-2 evolution should be acknowledged for future vaccination strategies that target most at-risk populations, especially if vaccination campaigns will cover a substantial part of the population. Mutations in immunologically-relevant genomic regions, viral recombination, virulence and fitness evolution must be considered when designing a future vaccination strategy. Finally, we would like to stress that despite potential implications of vaccination for evolution in the antibody epitopes, in face of an unprecedented global health crisis like the one we just experienced, mass vaccination is probably the only tool to prevent widespread loss of human lives and huge economic costs.

In my original review, I did not intend the authors to revise their manuscript to suggest that their approach is superior to work by Gog and others. Instead, they should do a much better job of acknowledging that their own work relies on assumptions that may not be valid (including uncertainty in the conceptual model, as well as its inputs) - and that this work is intended to demonstrate that alternative conclusions about the effects of vaccination can be reached using a different framework compared to earlier work (rather than arguing so strongly that the earlier work was too basic, and that the conclusions here are definitely correct).

We concluded the paragraph where we discuss work by Gog and others as follows (Lines 453-467):

From the evolutionary viewpoint, multi-locus models are closer to reality than independent-locus models. At the same time, based on strong genetic variation in the antibody epitopes in spike of SARS-CoV-2, we assumed that mutations in these regions have a low cost, by analogy with mutations in hemagglutinin protein of influenza. The validity of these assumptions remains to be tested directly in the future.

Since the authors have chosen not to present the possibility that a range of conclusions can be reached - and have instead chosen to argue fully for the results emerging from their model (however sophisticated it may be), I am afraid that I cannot support publication of this Perspective. As I said previously, a substantial change in the tone of the Perspective could certainly allow this work to be a nice addition to the literature.

As requested, we edited the text throughout to highlight that vaccination might potentially have impact on antigenic evolution as opposed to stating that vaccination does have impact on antigenic evolution. Substantial changes have been done to the abstract as well (Lines 23-31):

Mass vaccination was the main pillar of the public health response to the COVID-19 pandemic. It was very effective in reducing hospitalizations and deaths. At the same time, SARS-CoV-2 may escape from both natural and vaccine-induced immunity. We provide a perspective in the context of the viral evolutionary theory on how vaccination might accelerate SARS-CoV-2 evolution in antibody-binding regions compared to natural infection, at the population level. Using the evidence of strong genetic variation in antibody-binding regions and taking advantage of the similarity between the envelope proteins of SARS-CoV-2 and influenza, we assume that immune selection pressure acting on these regions of the two viruses is similar and discuss existing models of influenza evolution. We further discuss the implications of this phenomenon for future vaccination strategies.

Even the addition of a sentence expressing uncertainty at the end of the abstract would help (with similar phrasing changes later in the manuscript). For example, they could replace the last sentence of the abstract with: "Of course, this is purely theoretical and so this conclusion is far from certain. However, if it holds, we discuss the potential implications of this phenomenon for viral dynamics and future vaccination strategies."

We addressed this comment above.